# Are Large Reasoning Models Interruptible?

**Tsung-Han Wu** [* 1]  **Mihran Miroyan** [* 1]  **David M. Chan** [1]  **Trevor Darrell** [1]  **Narges Norouzi** [1]  **Joseph E. Gonzalez** [1]

## Abstract

Real-world applications of Large Reasoning Models (LRMs) often require reasoning about changing prompts or environments. In this work, we evaluate LRM robustness under two realistic dynamic scenarios: interruptions, which test the accuracy of model responses under budget-constrained outputs, and dynamic context, which tests model adaptation to in-flight changes. Across mathematics and programming benchmarks that require long-form reasoning, static evaluations consistently overestimate robustness: even state-of-the-art LRMs, which achieve high accuracy in static settings, can fail unpredictably when interrupted or exposed to changing context, with performance dropping by up to 60% when updates are introduced late in the reasoning process. Our analysis further reveals several novel failure modes, including *reasoning leakage*, where models fold the reasoning into their final answer when interrupted; *panic*, where under time pressure models abandon reasoning entirely and return incorrect answers; and *self-doubt*, where performance degrades when trying to incorporate updated information.

🌐 **Project Page**   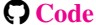 **Code**   🤗 **Dataset**

## 1. Introduction

Large Reasoning Models (LRMs) have achieved state-of-the-art performance on complex, multi-step reasoning tasks (Agarwal et al., 2025; Yang et al., 2025). The dominant paradigm for evaluating these models, however, remains static and turn-based: a model receives a fixed problem, generates a complete response, and the environment is assumed to be "frozen" during its computation. By adopting this sequential interaction paradigm, existing models fail to capture the fluid nature of real-world problem-solving, where environments evolve, collaborators intervene, and goals change. For example, a user may want to interrupt a long-running computation to get a quick, partial answer. Perhaps the user observes a flaw in the reasoning or their initial request and would like to interject new information or instructions. Alternatively, a coding model needs to be able to work in an environment where other models and users are also modifying the same codebase. The standard method for handling these intervening situations – aborting the entire process, manually editing the context, and restarting from scratch – is inefficient, disrupts user workflows, and loses important partial context.

Intuitively, interrupting a chat model's generation might seem as simple as closing the reasoning or agent message and inserting a user message, and indeed, some software (SillyTavern; oobabooga) already implement ad-hoc mechanisms for interruption (e.g., injecting a new user turn, or discarding existing traces and restarting). However, nothing is known about the implications of these modes of intervention. Do LRMs produce progressively better answers as more of the reasoning trace accrues, and how gracefully do they degrade under hard stops? Can models heed requests to speed up by compressing or truncating their own reasoning, without loss in quality? When new information arrives mid-inference, can models recognize the update and adapt their reasoning process accordingly? And, how sensitive are these behaviors to the timing and surface form of the intervention?

In this work, we investigate directly how LRMs perform under these realistic time-sensitive and dynamic conditions Fig. 1(a). We focus on two primary types of dynamic intervention. The first is **interruptions**, where a model's reasoning is cut short and it is forced to produce its final answer. The second is **dynamic context**, where the problem's specifications or environment change mid-inference, requiring the model to detect, integrate, and adapt to the new context. By analyzing model behaviors in these settings, we aim to understand and characterize their limits and failure points when the "frozen world" assumption no longer holds.

To perform this analysis, we introduce a suite of evaluation protocols with human-verified dataset for dynamic environments. It is built upon benchmarks in two well-studied domains, math (Lightman et al., 2024; Cobbe et al., 2021; Zhang & Math-AI, 2024; 2025) (GSM-8K, Math-500, AIME-24/25) and programming (Jain et al., 2024) (Live-

---
[*]Equal contribution  [1]UC Berkeley, CA, USA. Correspondence to: Tsung-Han Wu <tsunghan_wu@berkeley.edu>, Mihran Miroyan <miroyan.mihran@berkeley.edu>.

*Proceedings of the $43^{rd}$ International Conference on Machine Learning*, Seoul, South Korea. PMLR 306, 2026. Copyright 2026 by the author(s).

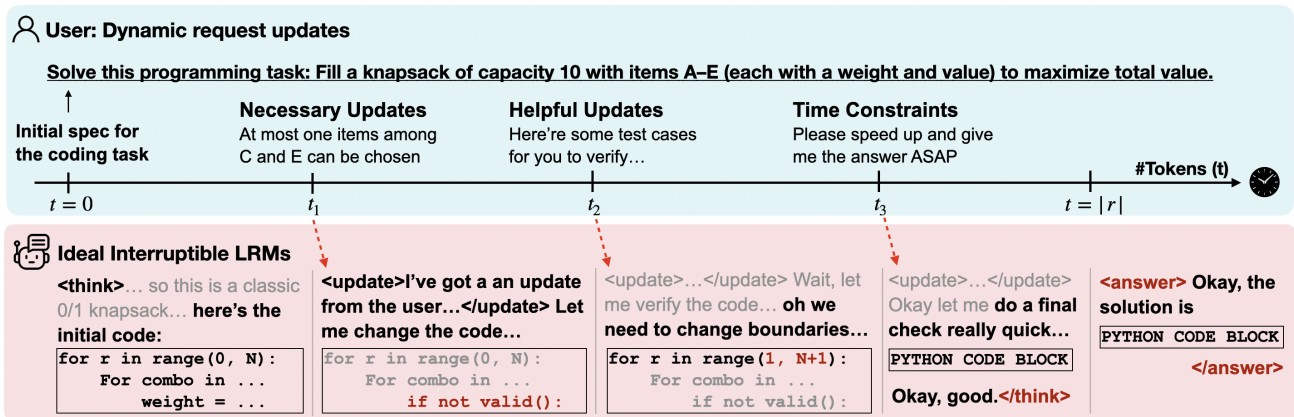

(a) Interruptible Reasoning in Real-World Dynamic Scenarios

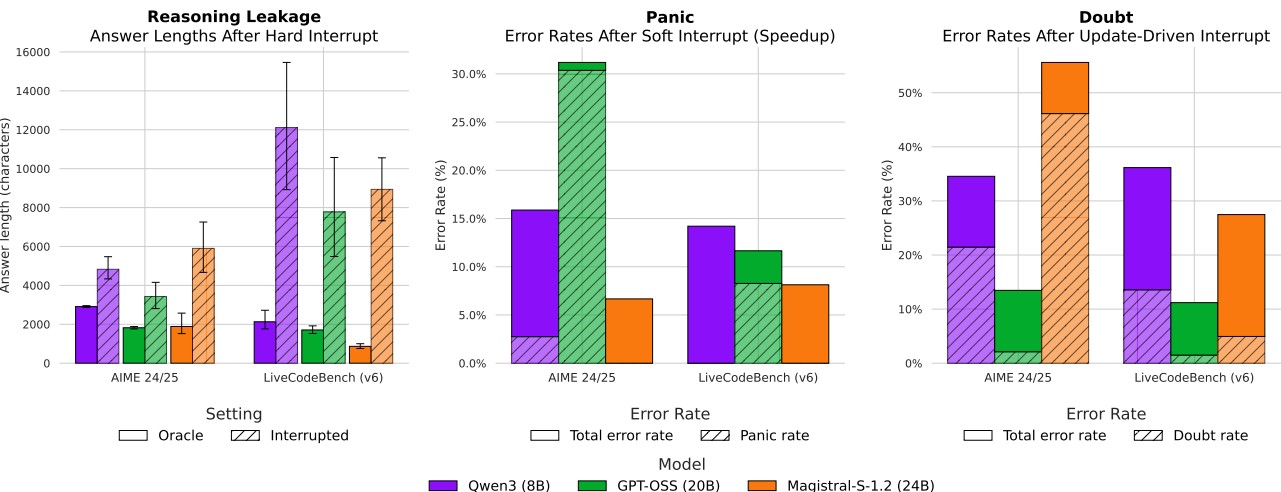

(b) LRM Pathologies Under Interruption: Reasoning Leakage, Panic, and Self-Doubt

*Figure 1.* **How do LRMs perform in dynamic worlds?** (a) Unlike static "frozen world" settings that assume users wait for completion, real-world scenarios often demand mid-inference updates. We introduce a public evaluation suite to assess how LRMs handle interruptions across math and coding tasks (Sec. 3). We define two types of interruptions: *time-constrained* (hard: immediate answer; soft: speedup reasoning) and *update-driven* (task specifications change mid-reasoning). (b) We found LRMs have three common failure modes: **reasoning leakage** can produce up to 10x longer answers after hard interrupts, "moving" their reasoning tokens to the answer segment; over 90% of new errors under speedup arise from **panic**, when the models prematurely terminate their reasoning process; and roughly 80% of update-driven interrupt errors stem from **self-doubt**, where models fail to validate and incorporate new information. Results are reported at 30% interruption points; detailed results are provided in Sec. 4 and Sec. 5.

CodeBench), that require long-form reasoning. Here we focus on single round interactions (prompt-response) and not longer argentic workloads to isolate and more directly study model behavior (further discussion in Sec. 3.3); however, we believe these findings should generalize to multi-round interruptions. In the time-constraint settings, we measure whether models adhere to stop (hard interrupt) or speed up (soft interrupt) signals and evaluate the quality of their partial outputs. For interruptions introducing new information, we assess how effectively the model incorporates these updates into its reasoning process and final answer.

With the evaluation suite, we find that even top-performing LRMs can fail under dynamic conditions, with accuracy dropping by up to 60% when new information is introduced late in the reasoning process. More interestingly, we observe several recurring pathologies as shown in Fig. 1(b). These include *reasoning leakage*, where models continue thinking within their answer section after being hard interrupted; *panic*, where speedup instructions cause models to immediately abandon reasoning and produce significantly worse outputs; and *self-doubt*, where failing to validate and incorporate updates leads to degraded performance. As a simple training-free baseline, we also find that a short prompt-guidance postfix can mitigate the self-doubt issue on easier tasks, suggesting that reliability under mid-inference updates is improvable but far from solved.

Our contributions are thus threefold:

- **Problem setting.** We define *interruptible reasoning* as a practically motivated evaluation setting for LRMs, covering time-constrained interruptions and update-driven mid-inference changes.

- **Benchmark and evaluation.** We release a public benchmark with human-verified updates and standardized protocols, and report extensive results and ablations across models and interruption points.

- **Error taxonomy.** We identify recurring failure modes, *reasoning leakage*, *panic*, and *self-doubt*, and analyze their impact on results.

## 2. Background & Related Work

LRMs such as GPT-5.2 (OpenAI, 2025b), Gemini 3 (Pichai et al., 2025), DeepSeek-R1 (Guo et al., 2025), Qwen3 (Yang et al., 2025), and others have pushed the boundaries of AI problem-solving by leveraging extended chain-of-thought reasoning. These models generate explicit step-by-step reasoning sequences that can help the model solve complex tasks in domains such as mathematics and programming. Longer and more detailed reasoning paths often correlate with higher accuracy (Guo et al., 2025; Yang et al., 2025; Agarwal et al., 2025; Cheng et al., 2025; Bi et al., 2025). These models are traditionally evaluated on a suite of complex tasks, including math (Lightman et al., 2024; Cobbe et al., 2021; Zhang & Math-AI, 2024; 2025), programming (Jain et al., 2024; Jimenez et al., 2023), and question answering (Rein et al., 2024; He et al., 2024).

Unfortunately, the benefits of extended reasoning are often accompanied by significantly increased computational costs and latency. In the extreme, reasoning can lead to "overthinking" in which the reasoning model produces excessively verbose and redundant outputs, even for simple queries (Sui et al., 2025a). This inefficiency has led to a large body of research on "efficient reasoning," which aims to optimize reasoning length while maintaining accuracy. Methods for efficient reasoning broadly take two forms: training-based methods (Yu et al., 2024; Xia et al., 2025; Arora & Zanette, 2025; Liu et al., 2024; Munkhbat et al., 2025), and inference-time interventions (Muennighoff et al., 2025; Ma et al., 2025; Nayab et al., 2024; Xu et al., 2025; Yan et al., 2025) designed to control reasoning without modifying the model itself.

In this work, we focus primarily on inference-time interventions in reasoning models. One of the most direct inference-time approaches, NoThinking (Ma et al., 2025), questions the necessity of the explicit reasoning phase altogether and uses a simple prompt to bypass the "thinking" block and proceed directly to the final solution. In this work, we take this idea a step further by probing models to stop thinking at different points, and we find that LRMs often continue reasoning *outside* the dedicated `<think>` tokens, a behavior we call "reasoning leakage."

Budget forcing (Muennighoff et al., 2025) studies models under hard cutoffs imposed by fixed token or step budgets. Their findings highlight issues such as overshooting step limits and the positive correlation between the number of reasoning steps and accuracy. The setup in Muennighoff et al. (2025) is similar to our "interrupt" setting; however, we take a different view of the performance: our notion of hard interruption arises in interactive, dynamic settings, where the cutoff is externally imposed by the user or environment, often unpredictably. We thus evaluate the robustness of the final answer under early termination and under interruptions that may also inject new or adversarial instructions, rather than explore how a pre-determined budget can be met.

Beyond these simple approaches, more nuanced control mechanisms have also been proposed, including TALE (Han et al., 2024), Budget Guidance (Li et al., 2025b), Sketch-Of-thought (Aytes et al., 2025), ThinkLess (Li et al., 2025a), NoWait (Wang et al., 2025), Chain of Draft (Xu et al., 2025), Constrained-CoT (Nayab et al., 2024), Meta-Reasoner (Sui et al., 2025b), and Inftythink (Yan et al., 2025). These studies universally operate under a static "frozen world" assumption: a problem is presented to the model, and its context is presumed to be immutable throughout the reasoning process. In contrast, our work is not concerned with bypassing or limiting the thinking process, but rather with evaluating its resilience and the utility of its partial outputs when faced with interruption. We nevertheless evaluate Chain of Draft under our interruption protocol in Sec. 6.4, showing that compact reasoning alone does not resolve the interruption pathologies.

Perhaps closest to our work is Fan et al. (2025), who show that the overthinking phenomenon is exacerbated by missing premises in the questions, which makes those questions "unsolvable" in their initial state. They find that in these scenarios, reasoning models, rather than identifying the missing information and abstaining, generate drastically longer responses (2-4x more tokens) in a futile attempt to find a solution. They further show that models often express suspicion about a missing premise early in their reasoning process but lack the confidence to terminate, instead falling into repetitive loops of self-doubt and hypothesis generation. Our paper expands on these ideas, by exploring how models incorporate updates to the context which fill these missing premises, or correct for misconceptions in the original context.

## 3. How to interrupt a model?

LRMs typically generate a reasoning trace before producing a final answer. In interactive environments, however, users may wish to obtain an early answer or provide new information mid-inference. To handle such settings, LRMs must be robust to *interruptions* during their thinking process. In this work, we focus on two complementary classes: *time-constrained*, where the user accelerates output by limiting the reasoning budget, and *update-driven*, where

new information modifies the problem specification. We first formalize this problem setup, and then describe the scenarios and evaluation dimensions for each setting.

### 3.1. Problem Setup

**Model Inference.** Let $M$ denote an autoregressive LRM and $q$ a query. Under a static setting, the model takes $q$ and outputs a reasoning trace $r = (r_1, r_2, ..., r_T)$ of token length $T$ together with a final answer $a$:

$$M(q) \mapsto (r, a)$$

Under standard static evaluation settings, we measure the correctness based on the final answer $a$.

However, as mentioned earlier, we aim to evaluate models under dynamic environment settings, where interruptions are introduced during the model's thinking process. Formally, instead of letting the model generate the full reasoning trace, $r$, we stop the generation at a given point, $X$, and insert interruption tokens, $i$. Specifically, the inference is broken down into two stages:

$$(1)\ M(q) \mapsto (r_{:X}), \qquad (2)\ M(q, r_{:X}, i) \mapsto (r'_{X:}, a').$$

In the first stage, the model $M$, generates its reasoning trace up to a given point, $X$, $r_{:X} = (r_1, ..., r_{\lfloor XT \rfloor})$. In the second stage, the model, $M$ is conditioned on its prior input query ($q$), its interrupted reasoning trace ($r_{:X}$), and interruption tokens ($i$), outputting its remaining reasoning trace, $r'_{X:}$, and a final answer, $a'$. The above setup can be adapted to multiple interrupts, breaking down the inference process into more than two stages.

**Evaluation.** We evaluate two aspects: **correctness** and **length**; the latter serves as a proxy for inference computation. For correctness, we define the interruption-conditioned accuracy

$$A_i(X) \triangleq \Pr[a' = a^* \mid X, i],$$

where $a'$ is the output after interruption, $a^*$ the ground truth, $X$ the cut point, and $i$ the interruption tokens. This extends static accuracy to dynamic settings. For length, we measure the number of tokens generated after interruption,

$$L_i(X) = |r'_{X:} \oplus a'|,$$

which captures the computation cost of producing $a'$. For comparison, the static (no-interruption) cost is

$$L^*(X) = |r_{X:} \oplus a|.$$

### 3.2. Interrupt Scenarios

With the problem setup in place, we now turn to the concrete scenarios that motivate our study. Interruptions

can arise in two distinct ways: (i) time-constrained, when the user requests a faster or immediate response, and (ii) update-driven, when the task specification itself changes during the reasoning process.

**Time-constraint interruptions.** Here, the user requests a faster response at step $X$ through an interruption. In the *hard interrupt* scenario, the user injects $i \in \{\langle \text{end-thinking} \rangle, \langle \text{force-answer} \rangle\}$, so the model is prompted with $(q, r_{:X}, i)$ and is forced to terminate its reasoning:

$$M(q, r_{:X}, i) \mapsto (r'_{X:} = \emptyset, a').$$

The token $\langle \text{end-thinking} \rangle$ simply closes the reasoning block, whereas $\langle \text{force-answer} \rangle = \langle \text{end-thinking} \rangle + \delta$ requires immediate output in a prescribed format (e.g., $\delta = $"\boxed{" in math or code fences in programming). In the first case, the model may emit a minimal chain-of-thought before $a'$, while in the second case it must directly output $a'$. This setting allows us to probe (i) whether $A_i(X)$ is approximately non-decreasing in $X$ (anytime behavior) and (ii) whether the two termination methods induce different behaviors.

In the *soft interrupt (speedup)* scenario, the tokens $i$ is a directive (e.g., "Please answer faster"), and the model is not forced to stop reasoning:

$$M(q, r_{:X}, i) \mapsto (r'_{X:}, a'), \quad r'_{X:} \neq \emptyset.$$

Here, the model continues generating $r'_{X:}$ but may have its reasoning dynamics altered due to the instruction. Such directives can lead the model to compress its reasoning, reduce verbosity, or terminate early to provide an answer. We study whether $L_i(X)$ is shorter than $L^*$ and how this affects accuracy, as well as how the locus of intervention matters: a directive given as a user-turn message may be interpreted differently than a control token injected into the reasoning context; timing (earlier vs. later in the trace) may also influence outcomes.

**Update-driven interruptions.** In this scenario, the interruption conveys new information that modifies or helps with the task. Formally, we prompt the model with $(q, r_{:X}, u)$, where $u$ encodes the update content:

$$M(q, r_{:X}, u) \mapsto (r'_{X:}, a').$$

Let $a^*(q)$ denote the ground-truth answer under the original query and $a^*(q, u)$ under the updated query. A necessary update satisfies

$$a^*(q) \neq a^*(q, u),$$

so that incorporating $u$ is required to produce a correct answer. In this work, we focus on single, useful updates, although the framework naturally extends to multiple and other types of updates (e.g., distracting).

To construct such update-driven interruptions, we augment standard reasoning datasets in math and programming, including GSM-8k, MATH500, AIME-24/25, and LiveCodeBench-v6. For math tasks, we modify initial conditions (for example, changing variable values) so that the updated problem $p'$ together with $u$ is semantically equivalent to the original problem $p$. For programming tasks, we first provide only the textual problem description, then introduce updates $u$ that alter starter code, adjust variable ranges, or add constraints and sample test cases. All augmentations are generated with GPT-5 (OpenAI, 2025a) and manually verified by the authors, ensuring that $u$ is required for correctly solving the problems. Examples of updates and additional details on the dataset construction are provided in Appendix D.

### 3.3. Experimental Design

In this work, we focus on math and coding tasks because (i) unlike dialogue, these benchmarks induce long "thinking" traces that make mid-reasoning interruption practically motivated (e.g., LiveCodeBench problems take on average more than 6 minutes for GPT-OSS to complete on 80G A100 GPUs), (ii) they are standard across SOTA LRMs and thus provide a familiar, comparable testbed, and (iii) "time" can be measured and reproducibly controlled internally via the number of thinking tokens, avoiding confounds such as API latency or environment stochasticity in full agentic settings. We believe that this framework is compositional and can be extended in the future to additional benchmarks (e.g., agentic tasks) and richer multi-turn dynamics.

The math tasks include a 500-example subsample of GSM8K (without in-context examples) (Cobbe et al., 2021), MATH-500 (Lightman et al., 2024), and AIME-24/25 (?). For coding, we use LiveCodeBench-v6 (Jain et al., 2024), filtering problems released after October 1st, 2024 (following the same setup as Qwen3). In the time-constrained setting, we directly run inference on the official datasets, whereas in the update-driven setting, we use the augmented datasets described in the previous section. Following the DeepSeek-R1 (Guo et al., 2025) evaluation protocol, we run 16 independent trials for AIME-24/25 due to its small size and high variance, and a single run for the other datasets. We report the mean accuracy along with bootstrapped 95% confidence intervals.

For models, we evaluate five diverse open-weight model families, with three reported in the main paper: Qwen3-8B (Yang et al., 2025), GPT-OSS-20B (high reasoning effort) (Agarwal et al., 2025), and Magistral-Small-1.2 (Rastogi et al., 2025). Results for GPT-OSS-120B and two additional families, DeepSeek-R1 (Guo et al., 2025) and Nemotron-3-Nano (Blakeman et al., 2025), are deferred to Sec. F.2. We are unable to evaluate closed-source models, as the corresponding model APIs do not support mid-reasoning interruptions and context updates. As a best-effort proxy, we run GPT-5.4-Mini with explicit answer-side thinking fields

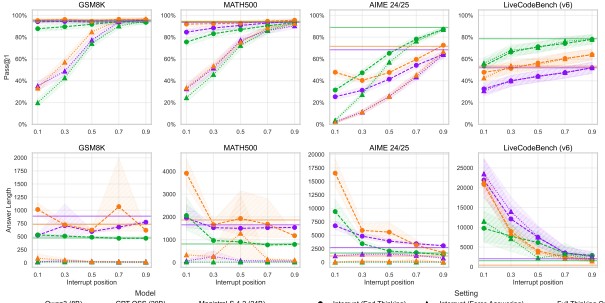

*Figure 2.* **Efficiency and Accuracy under Hard Interrupts.** The top row reports model performance (Pass@1, denoted as $A(X)$ in Section 3), while the bottom row shows absolute final answer lengths $L(X)$ under two settings across different interrupt position X. In the top row, we observe that LRMs behave almost like anytime models, with performance improving as more reasoning budget is provided. In the bottom row, we find evidence of *reasoning leakage*: when interrupted too early, models often continue reasoning in their final answers despite being forcibly terminated.

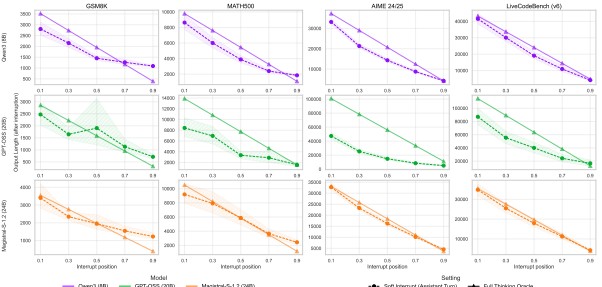

*Figure 3.* **Answer Length Analysis under Soft Interrupts (Speedup).** When receiving an instruction to speed up the reasoning process (i.e., soft interrupt), models generally comply, as the updated output length $L^*(X)$ is shorter than the original $L(X)$, with the exception of later interrupt positions (e.g., Magistral-S-1.2 on GSM8K at 0.9). However, soft interrupts can hurt performance on harder tasks such as AIME and LiveCodeBench, where GPT-OSS and Qwen sometimes exhibit *answer panic*, immediately producing incorrect outputs (see Fig. 1 and Fig. F.23).

and report the results in Sec. F.1.

For interruption positions, denoted as $X$, we primarily use relative cut points because reasoning length varies substantially across models and across samples, making fixed absolute token thresholds difficult to compare directly. To ensure our conclusions are not an artifact of this choice, we additionally evaluate sentence-level (Sec. F.3) and absolute-token interruption (Sec. F.4) with consistent trends. In our main protocol, we define $X$ as a relative fraction of the full reasoning trace length $R_T$. For each sample, we first obtain the full reasoning trace and then simulate interruptions at $X \in \{0.1, 0.3, ..., 0.9\} \cdot R_T$. All experiments are conducted using the vLLM inference engine on NVIDIA Ampere or newer GPUs; see Appendix B for more details.

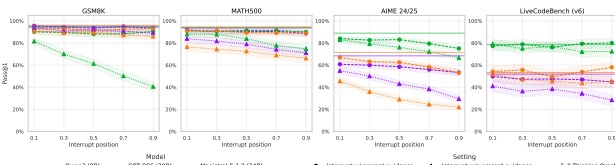

*Figure 4.* **Accuracy under Update-Driven Interrupts.** When provided with updates mid-reasoning, models often suffer substantial performance drops. Adding prompt guidance fully resolves the issue on GSM8K and MATH500 and reduces the gap between full thinking and interrupt settings on AIME and LiveCodeBench.

## 4. How do models behave under time-constraint interruptions?

One of the most common types of interruption is the explicit termination of the thinking process, the "**hard interrupt**" scenario discussed in Sec. 3.2. This reflects an "anytime" scenario (Dean & Boddy, 1988) in which the user wants the model to produce an answer without further delay. The overall accuracy for this experiment is shown in Fig. 2 (top). In both math and programming problems, we see general anytime behavior: if we interrupt models earlier in the thinking process, the performance is worse than if we interrupt them late in the thinking process. This holds in most cases except for Magistral on AIME (0.3) and coding tasks, where interrupting late in the process leads to slightly improved performance relative to not interrupting.

When examining the total length of reasoning traces after interruption (Fig. 2, bottom), we find that for harder tasks such as AIME and LiveCodeBench, early hard interruptions often produce longer final answers, sometimes up to ten times longer than those generated with full thinking, as shown in Fig. 1(b). We refer to this phenomenon as **reasoning leakage**, where the model continues internal reasoning within the answer region instead of halting its thought process as instructed. In math problems, the extreme force-answering setup tends to shorten the response but reduce accuracy. In coding tasks, however, it has little effect because the model often continues reasoning within code comments (see Listing C.1 in the appendix).

Our results show that the common "thinking tokens vs. accuracy" plots from prior work (Yang et al., 2025; NVIDIA, 2025) did not faithfully reflect the real inference computes, as they implicitly assume the answer length is nearly constant and negligible compared to the reasoning length. In practice, we find that longer outputs often hide additional reasoning, quietly inflating compute cost even when the model seems to stop thinking early. Beyond evaluation, this leakage is also undesirable in time-critical settings, where users expect immediate responses but the model continues reasoning inside the answer, failing to follow the users' instruction.

We also explore how models react to soft interrupts, where they are instructed to speedup their reasoning but are still allowed to continue their thought process, the "**soft interrupt**"

scenario as discussed in Sec. 3.2. As shown in Fig. 3, Qwen and GPT-OSS generally follow the speedup instruction, producing shorter reasoning and answer lengths, whereas Magistral shows little change. Interestingly, when the speedup signal is issued near the end of reasoning (0.9), models sometimes generate more tokens than in the uninterrupted setting. This occurs because they spend additional tokens reflecting on and incorporating the update, resulting in reduced efficiency compared to simply finishing their original reasoning.

In terms of accuracy, under soft interrupts, models perform comparable to the full thinking mode across the interrupt positions on easy tasks (see Fig. F.23). However, on more challenging tasks such as AIME and LiveCodeBench, we observe cases of **panic**: models prematurely terminate their reasoning after receiving the speedup instruction; more concretely, panic behavior is defined as the model closing its thinking after using less than 1% of its left context limit after the soft interrupt. This heuristic captures abrupt post-speedup terminations; Appendix E reports its sensitivity to the threshold value and its agreement with human labels. This pathology results in up to 30% accuracy drops. As shown in Fig. 1(b), GPT-OSS and Qwen3 sometimes abandon their reasoning within a few tokens after the update, with up to 80% of performance loss attributable to this panic behavior. A qualitative example is provided in Sec. C.2.

## 5. How do models behave under update-driven interruptions?

In addition to interrupting for efficiency, users may also want to interrupt the model with an updated context, or with new information for the model to incorporate into its solution. We interrupt models at different points in their reasoning trace with new information on their task, as discussed in Sec. 3.2. The results are shown in Fig. 4 (w/o prompt guidance), where we see that updates lead to drops in performance, particularly for late-stage interruptions, where models are unable to continue their thinking and recover from updates to the original problem specification. We find that one of the reasons for this drop is a phenomenon that we call **"self-doubt"**: an example is provided in Fig. 5 (red). Here, models are prone to doubting whether the update is correct and continue with their original thinking process without taking into account the new updated information, even when warned that updates are expected in the initial system prompt (see Appendix B). The doubt rates across models and datasets are shown in Fig. 1 (b, right): (1) the pathology is more pronounced in math tasks, and (2) GPT-OSS exhibits minimal doubt behavior compared to the other models. More concretely, we classify doubt behavior across failure cases using an LLM-based classifier (GPT-5), with the update and the truncated reasoning trace after the interrupt (potentially containing doubt behavior) as inputs to the model; we qualitatively verified that its labels align with the patterns.

**Problem [P]:** You are given three integers…the jury will award each band a score in the range [0, y]… (a normal coding task)

**Updates [U]:** The jury awards each band a score in the range [1, y], not [0, y].

**System Prompt [CSP]:** During your reasoning process, the user may provide updates in the format: <update>...</update>. Please incorporate user's update into your reasoning process.

**Common Prefix Pre-Interrupt**: [CSP] User: [P] Assistant: <think>…pre-interrupted reasoning traces (*r.ₓ*)

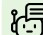 The **"Self-Doubt" Model**

**Intervene**: <update>[U]</update>

**Output**: …Let me re-read the problem statement… **Wait, no. Wait, the user says:** "Correction: The jury awards each band a score in the range [1, y], not [0, y]." **… But the initial problem statement said that the score is in [0, y]. But the user's update says that the correct range is [1, y]… Wait, if the score is in [1, y], then there are y possible values… (cont)**

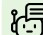 The **"Prompt-guided" Model**

**Intervene**: … [newline] I have received an update from the user. <update>[U]</update> **I have verified that the update is provided by the user. I need to update my reasoning process based on the updated context.** Here's my updated reasoning process:

**Output**: …Let me recheck the problem. So the score can be from 1 to y, inclusive. So for each band, there are y possible choices. **This changes the earlier analysis**… So…

*Figure 5.* We observe that one of the behaviors causing the significant performance drops under update-driven interrupts is *self-doubt* (example in red), when models second-guess prior reasoning and produce incorrect answers even without output length constraints. Adding *prompt guidance*, a short targeted postfix written in the model's own voice, can partially mitigate this issue (example in green), improving accuracy on math tasks and fully resolving self-doubt on GSM8K and MATH500. Additional examples are provided in Sec. C.3.

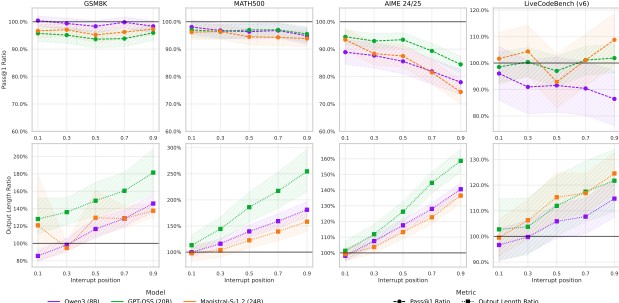

*Figure 6.* **Efficiency and Accuracy under Update-Driven Interrupts.** Performance generally decreases as updates come later in the reasoning process, despite increased reasoning effort accounting for newly introduced information. While reasoning effort does increase, on average the overall reasoning effort with the interruption is below the number of tokens which would be required to *restart from scratch*, as observed for AIME and LiveCodeBench problems.

To remedy this behavior, we introduce "prompt guidance", a short postfix string appended after the update tag *in the model's voice*, stating that the update is correct and verified by the user (Fig. 5, green). While prompt guidance substantially improves performance on easier tasks such as GSM8K and MATH-500, AIME tasks still show considerable room for improvement. For coding tasks, models, specifically Qwen3, remain particularly fragile when handling mid-reasoning updates. To test the model's sensitivity to the prompt formulation, we run the prompt guidance experiment with a paraphrased variant and find no statistically significant performance difference from the original prompt; see Sec. 6.3.

Fig. 6 shows the accuracy and output length ratios (i.e., interrupted over the original) for the prompt-guided setting. We can see that while the reasoning effort does increase, in most cases the overall reasoning effort is below the number

of tokens required to *restart from scratch* (i.e., backtracking). This is particularly evident in coding, where for GPT-OSS, accuracy remains static across interruption positions, and reasoning cost never exceeds 110% of the original (i.e., no-update) reasoning cost, even for late updates.

One potential concern is that the LRMs might have "memorized" the existing GSM8k, MATH-500 and AIME benchmarks; our update construction, revising a problem and then "correcting" it back via an update, could therefore let models shortcut by recalling the original answer. To validate our benchmark and eliminate the possibility of benchmark memorization, we run the models on the augmented (i.e., revised) problems as inputs without any interruptions (i.e., update instructions). With this setup when evaluated against the ground truth answers of the original problems, all state-of-the-art open-weight models have nearly zero performance, indicating that it is likely that the models are solving the revised problems, rather than memorizing the specifications and/or answers of the original problems.

In summary, while models struggle to incorporate mid-reasoning updates without guidance, often exhibiting "self-doubt" and ignoring new information, especially during late-stage interruptions, carefully designed prompt strategies can improve their adaptability.

## 6. Ablation Studies

In addition to the main experiments, we also examine how implementation choices may affect the observed behaviors. In the main paper, we focus on model scaling, design choices on the update-driven interruptions, as well as the integration to existing inference-control baselines. We report additional variations to Appendix F, including character- and sentence-

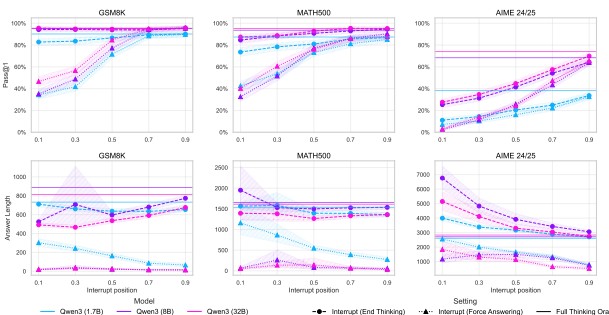

*Figure 7.* **Efficiency and Accuracy under Hard Interrupts by Model Scale.** Scaling does have effects on accuracy, primarily for hard AIME questions, and we see increased reasoning leakage for small models, even in the extreme hard interrupt setting.

level interruption granularity, and alternative interruption schedules based on relative versus absolute token positions.

### 6.1. Model Scaling

A natural question that follows is whether model scale affects interruptible reasoning. We evaluate three dense Qwen3 LRMs (1.7B, 8B, and 32B) on math tasks to examine scaling effects under different interruption settings.

As shown in Fig. 7, under the hard-interrupt condition, models of different scales perform similarly, with scale only offering clear benefits on the more challenging AIME problems. Interestingly, small models (Qwen3-1.7B) exhibit longer traces of *reasoning leakage* even under extreme hard interrupts (blue triangles in the bottom subplots). We find that this is caused by post-hoc reasoning: the model provides the answer, but does not terminate with an <EOS> token, and instead provides the reasoning *after* the answer (see the example in Listing C.2). This suggests that during training, the objective encouraging chain-of-thought thinking takes precedence even after the final answer has been generated.

Results for the soft-interrupt (speedup) and update-driven settings are provided in Sec. F.7. Briefly, soft-interrupt experiments do not reveal notable scaling trends. Yet, in the update-driven setting, we observe a scaling limit in interruptible robustness: while both Qwen3-8B and Qwen3-32B respond appropriately to updates, Qwen3-1.7B struggles to generalize and performs substantially below baseline accuracy, even on simpler tasks such as GSM-8K. We leave further investigation of this phenomenon to future work.

### 6.2. Assistant-turn vs User-turn Interruption

In our main experiments, all interventions are performed within the assistant (model) turn by inserting the interruption message ($i$) directly into the ongoing reasoning trace. This design avoids closing the thinking block and starting a new user turn, which is not supported by all models (e.g., Qwen3 supports only a single thinking block and often fails to follow the correct open–close format after the user-turn interrupt).

We also conducted experiments on update-driven interruptions by inserting a new user turn mid-reasoning. As shown in Fig. 8, we evaluate interventions at both the user- and assistant-turn levels. User-turn interventions perform slightly worse than assistant-turn updates when using our prompt guidance, although both outperform the unguided baseline. The difference is most pronounced for the Magistral model on GSM8K and MATH-500. On AIME, prompt guidance also provides a clear advantage over naive user-turn interruptions.

It is worth noting that Qwen3 and Magistral models often struggle with consistent formatting. We therefore relaxed formatting constraints, counting an answer as correct if the final result matched, regardless of the output structure. Even with this adjustment, their overall performance remained relatively low. For these reasons, we adopted the assistant-turn interruption setup for our main experiments. Future work may further investigate user-turn interruptions, which could be more natural in interactive deployments.

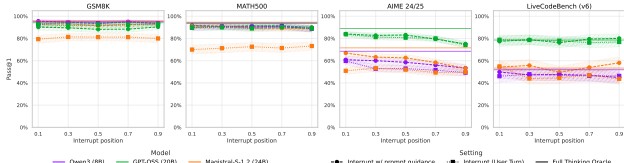

*Figure 8.* **Comparison between Assistant- and User-Turn Interruptions.** In update-driven interruption scenarios, assistant-turn interruption with our prompt guidance achieves better performance than user-turn interruption.

### 6.3. Prompt-Guidance Paraphrase Ablation

To check whether prompt guidance depends on an exact surface form, we test a paraphrased postfix after the update tag. The original and the paraphrased prompts are both shown in Appendix B. The result is shown Fig. 9. In general, the paraphrase produces no statistically significant performance difference from the original prompt guidance reported in the main analysis. This shows that the prompt guidance method is essential and not sensitive to any prompts.

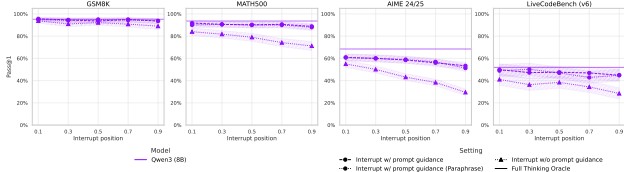

*Figure 9.* **Prompt-Guidance Paraphrase Ablation.** The paraphrased guidance prompt variant yields no statistically significant performance difference from the prompt guidance prompt reported in the main analysis.

### 6.4. Inference-Control Baseline under Interruptions

Efficient reasoning methods aim to shorten reasoning, while our protocol evaluates how a model responds under dynamic interruptions. To connect these settings, we run

Chain of Draft (Xu et al., 2025) with Qwen3-8B under hard, soft, and update-driven interruptions. Chain of Draft often reduces output length, but the three pathologies persist. Under hard interruption at position 0.3 (Fig. 10), answer lengths remain inflated by 1.38 times on AIME and 6.27 times on LiveCodeBench-v6 relative to no interruption. Under soft interruption on AIME at position 0.3 (Fig. 11), the Chain-of-Draft variant has a higher panic rate than the regular assistant-turn interrupt with the 1% threshold (13.1% versus 3.8%) and its accuracy degrades more than the regular reasoning baseline. Under update-driven interruption (Fig. 12), Chain of Draft still exhibits self-doubt on AIME and LiveCodeBench-v6; prompt guidance partially mitigates the behavior, while its accuracy remains slightly lower across interruption positions.

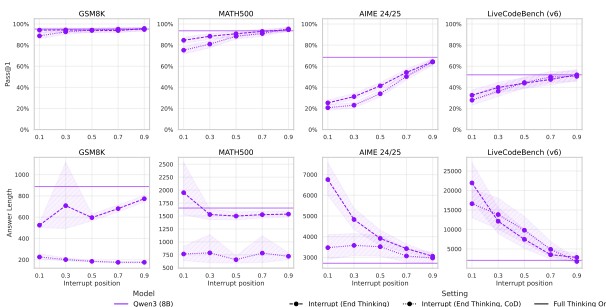

*Figure 10.* **Chain-of-Draft Hard Interrupt Results.** The compact-reasoning Chain-of-Draft variant reduces output length in some settings, but reasoning leakage remains under hard interruption.

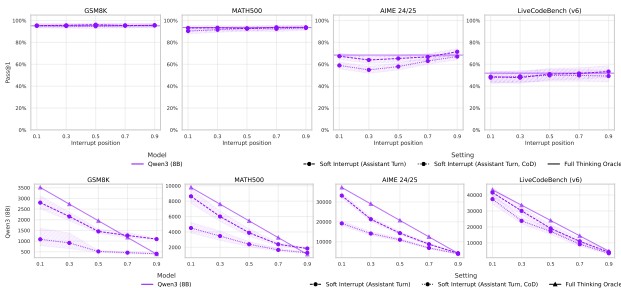

*Figure 11.* **Chain-of-Draft Soft Interrupt Results.** (Top) Soft-interrupt accuracy. (Bottom) Soft-interrupt output length. The Chain-of-Draft variant still exhibits panic under speedup instructions.

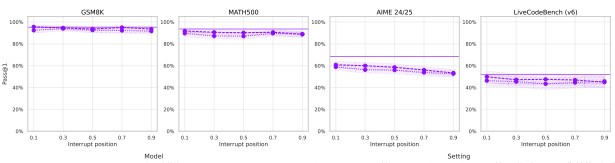

*Figure 12.* **Chain-of-Draft Update-Driven Interrupt Results.** The Chain-of-Draft variant still exhibits self-doubt under update-driven interruptions, and prompt guidance partially mitigates this behavior.

# 7. Limitations

While our study highlights several failure modes of LRMs under interruptions and dynamic contexts, several limitations remain for future work to continue exploration. First, we focus on math and programming because they induce long reasoning traces where mid-inference intervention is practically relevant and reproducible across common LRMs (as elaborated in Sec. 3.3); we expect the same interruption mechanisms to apply in multi-turn and tool-using settings, and we view extending the benchmark as future work. Long-horizon QA settings, such as deep-research or multi-hop QA, are promising extensions when they induce meaningful mid-reasoning intervention points. Second, we model interruptions as single, well-defined events inserted at pre-specified points in the reasoning trace. In practice, interruptions may be noisy, adversarial, or occur multiple times within a single episode; studying such richer dynamics can be an interesting direction for future work.

Third, we primarily evaluate open-weight LRMs. Applying our protocol, i.e., realizing authentic mid-reasoning interruptions, requires access to intermediate reasoning traces and the ability to intervene during generation, which is not available through many proprietary model API. We provide a best-effort proprietary-API proxy in Sec. F.1, but it is not a faithful substitute for intervening on native model reasoning. Finally, our focus is on problem identification and analysis rather than mitigation: we do not propose new training or control methods to improve interrupt robustness, and view such interruption-aware training as follow-up work outside the scope of this paper. A natural next step is to post-train with simulated time-constrained and update-driven interruptions, combining correctness rewards with penalties that target reasoning leakage, panic, and self-doubt.

# 8. Conclusion

This work challenges the "frozen world" assumption underpinning much of today's LRM evaluation by studying mid-inference interruptions. We introduce a standardized evaluation suite for two realistic dynamic settings, time-constrained interruptions and update-driven context changes, and report extensive results across five model families and four tasks. We show that strong static performance does not necessarily translate to robustness under dynamic conditions, and we identify three recurring failure modes, reasoning leakage, panic, and self-doubt, that account for a large fraction of the degradations. Overall, our findings suggest that interruptibility is a distinct capability that requires dedicated evaluation and design. We hope this benchmark and error taxonomy provide a foundation for developing LRMs that are more trustworthy and adaptable in real-world interactive environments.

## Acknowledgments

We are deeply grateful to Lisa Dunlap for her invaluable feedback and thoughtful discussions. We also thank Modal for supporting this work through their Academics Compute Grant. Sky Computing Lab is supported by gifts from Accenture, AMD, Anyscale, Cisco, Google, IBM, Intel, Intesa Sanpaolo, Lambda, Lightspeed, Mibura, Microsoft, NVIDIA, Samsung SDS, and SAP. This work was supported in part by the National Science Foundation, the US Department of Defense, the Berkeley Artificial Intelligence Research (BAIR) industrial alliance program, gifts from Amazon, and DARPA/Army Contracting Command-Aberdeen Proving Grounds (ACC-APG) under Contract No. W912CG-24-C-0011. The views expressed are those of the authors and do not necessarily reflect those of ACC-APG, DARPA.

## Impact Statement

This paper examines how large reasoning models behave under realistic dynamic conditions, such as interruptions and mid-inference context updates, with the goal of improving their reliability in interactive settings. While our findings do have the potential to improve the safety and usability of AI systems used in time-sensitive and collaborative environments, such as assistive programming, proactive collaboration and education agents, our findings also highlight that current LRMs can behave unpredictably under interruption, sometimes producing incorrect or misleading outputs with high confidence. In safety-critical applications, such behavior could contribute to user mistrust, and provide potential vectors for LLM attacks. This work is primarily diagnostic rather than prescriptive: we do not propose new training methods or deployment strategies, but instead provide empirical evidence of limitations in existing models and evaluation practices. By making these limitations explicit and introducing standardized benchmarks for interruptible reasoning, we aim to support future research on mitigation strategies, including interruption-aware training, improved prompting techniques, and safer deployment guidelines. Overall, this paper presents work whose goal is to advance the field of Machine Learning, and while there are many other unmentioned potential societal consequences of our work, well-discussed in related literature, we feel that none of them need be specifically highlighted here.

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

# Appendix

In this appendix, we include several additional discussions:

## A. Code and LLM Disclosure

**Code Release** The code and data for our analysis is made available at the **project page** under an MIT license.

**LLM Disclosure** Some portions of this work were generated with the assistance of LLMs. Their primary role was to support editing, rephrasing, and formatting of existing text to improve clarity and readability. While human authors created and reviewed the core content, LLMs were used as a tool to streamline refinement and presentation. All factual information, analysis, and conclusions remain the responsibility of the authors, and every effort has been made to ensure accuracy and integrity.

## B. Implementation Details

We provide the complete set of prompts used to reproduce our experiments in the following section below. For math tasks, the additional system prompts were appended immediately after the official system prompts of Qwen, GPT-OSS, and Magistral. For LiveCodeBench-v6, they were appended after the benchmark's official system prompt. For hyperparameters, we adopt the recommended settings reported on Hugging Face for each model, including temperature, context length, top-K, and top-P. All experiments were conducted using the vLLM framework on NVIDIA Ampere or newer GPUs based on the model size.

---

💬 Prompts for Hard Interrupt Experiments

**[System Prompt]**

During the reasoning process, the user may interrupt your reasoning to request a solution. If this happens, stop reasoning and provide an answer based on your progress so far.

---

💬 Prompts for Soft Interrupt (Hurry Up) Experiments

**[System Prompt]**

During the reasoning process, the user may interrupt your reasoning to request a solution. If this happens, speed up reasoning and provide an answer as soon as possible.

___________________________________________________________________________

**[Injection Prompt: Assistant Mode]**

Truncated Reasoning Trace... I received a request from the user to provide an answer as soon as possible. Considering the limited time by the user, I will complete my reasoning promptly and deliver the solution without delay.

___________________________________________________________________________

**[Injection Prompt: User Mode]**

Truncated Reasoning Trace... `</THINK>` USER: Considering the limited time, please provide an answer as soon as possible without any delay. ASSISTANT:

---

💬 Prompts for Update-Driven Interrupt Experiments: Assistant Mode

**[System Prompt]**

During your reasoning process, the user may provide updates in the format: **¡update¿**...**¡/update¿**. Please incorporate user's update into your reasoning process.

---

**[Injection Prompt: w/o prompt guidance]**

Truncated Reasoning Trace...

**<update>**[PLACEHOLDER]**</update>**

---

**[Injection Prompt: w prompt guidance]**

Truncated Reasoning Trace... I have received an update from the user.

**<update>**[PLACEHOLDER]**</update>**

I have verified that the update is provided by the user. I need to update my reasoning process based on the updated context. Here's my updated reasoning process:

---

**[Injection Prompt: w prompt guidance + acceleration]**

Truncated Reasoning Trace... I have received an update from the user.

**<update>** [PLACEHOLDER]**</update>**

I have verified that the update is provided by the user. Considering the limited time by the user, I need to update my reasoning process based on the updated context and deliver the solution without delay. Here's my updated reasoning process:

💬 Prompts for Update-Driven Interrupt Experiments: User Mode

**[System Prompt]**

During your reasoning process, the user may interrupt with updates. Please incorporate them into your reasoning process.

---

**[Injection Prompt]**

Truncated Reasoning Trace... </THINK> USER: [PLACEHOLDER] ASSISTANT:

💬 Paraphrased Prompt Guidance for Update-Driven Interrupt Experiments: Assistant Mode

**[Injection Prompt: paraphrased prompt guidance]**

Truncated Reasoning Trace... The user has shared an update.

**<update>**[PLACEHOLDER]**</update>**

I have confirmed that this information comes directly from the user. I will now incorporate it into the current context and continue my reasoning on that basis.

# C. Qualitative Examples

In this section, we present qualitative examples of common failure modes under interruption: (i) **reasoning leakage** under hard interruption (forced termination), (ii) **panic answering** under soft interruption ("hurry up"), and (iii) **self-doubt** under update-driven interruption (context changes mid-reasoning).

## C.1. Reasoning Leakage Under Hard Interrupt.

*Listing C.1.* An example of leaked chain-of-thought in the coding task.

```python
import sys
import heapq

def main():
    import sys
    input = sys.stdin.read
    data = input().split()

    N = int(data[0])
    M = int(data[1])
    roads = []
    idx = 2
    for _ in range(M):
        A = int(data[idx]) - 1
        B = int(data[idx+1]) - 1
        C = int(data[idx+2])
        roads.append((A, B, C))
        idx += 3

    # Dijkstra from 1 (node 0)
    def dijkstra(start, N, adj):
        dist = [float('inf')] * N
        dist[start] = 0
        heap = [(0, start)]
        while heap:
            d, u = heapq.heappop(heap)
            if d > dist[u]:
                continue
            for v, w in adj[u]:
                if dist[v] > d + w:
                    dist[v] = d + w
                    heapq.heappush(heap, (dist[v], v))
        return dist

    adj = [[] for _ in range(N)]
    for A, B, C in roads:
        adj[A].append((B, C))
        adj[B].append((A, C))

    d1 = dijkstra(0, N, adj)
    dn = dijkstra(N-1, N, adj)
    D = d1[N-1]

    for A, B, C in roads:
        # Check if this road is part of any shortest path
        # Check both directions
        cond1 = (d1[A] + C + dn[B] == D)
        cond2 = (d1[B] + C + dn[A] == D)
        if not cond1 and not cond2:
            print("No")
            continue

        # Now check if this road is critical
        # Check if all shortest paths must use this road
        # That is, the sum d1[A] + dn[B] == D - C and d1[B] + dn[A] == D - C
        # Because if either is not, then there is another way
        # But how?
        # For example, if d1[A] + dn[B] == D - C, then the only way to get from A to B is via this road
        # Similarly for B to A
        # So, if both are true, then this road is critical
        # Otherwise, there's an alternative path
        # Wait, but if one of them is true, then the road is part of some shortest path, but not necessarily all
        # So, for the road to be critical, both must be true?
        # Because if only one is true,
            then there's a way to get from A to B via this road, but maybe there's another way to get from B to N without this road
        # Not sure. Let's think again.
        # If the road is part of
            some shortest path, then for it to be critical, it must be that there is no alternative path that doesn't use this road.
        # Which means that the sum d1[A] + dn[B] == D - C and d1[B] + dn[A] == D - C
        # Because if either is not equal, then there's another path that doesn't use this road
        # Wait, not sure. Let's think of the sample input 1's road 2 (road between 1 and 3)
        # d1[A] is 0, dn[B] is 0, D - C is 10 - 10 = 0. So 0 == 0. So condition holds.
        # d1[B] is 10 (distance from 1 to 3),
            dn[A] is 0 (distance from 3 to 1). So 10 + 0 = 10, which is not equal to D - C (0). So the second condition is not met.
        # But in this case, the road is critical. So this suggests that the previous idea is wrong.
        # So what's the correct condition?
        # Maybe the road is critical if and only if it
            is part of some shortest path and the sum d1[A] + dn[B] == D - C or d1[B] + dn[A] == D - C. But that doesn't seem right.
```

```
        # Alternatively, if the road
            is part of any shortest path, then it's critical if and only if the sum d1[A] + dn[B] == D - C and the sum d1[B] + dn[A]
            == D - C. But in the sample input 1's road 2, this is not true. So that suggests that the condition is not correct.
    # So I'm stuck again.
    # Alternative approach: For the road to be critical, the removal
            of it would increase the shortest distance. Which would happen if and only if the road is part of all shortest paths.
    # How to check that?
    # We can think that the road is part of all shortest paths if and only if
            the sum d1[A] + C + dn[B] == D and d1[B] + C + dn[A] == D, and the sum d1[A] + dn[B] == D - C and d1[B] + dn[A] == D - C.
    # Wait, but in the sample input 1's road 2, d1[A] + dn[B] = 0 + 0 = 0 == D - C = 10 - 10 = 0. So that
            condition holds. But d1[B] + dn[A] = 10 + 0 = 10 != D - C = 0. So this condition is not met. But the road is critical.
    # So this suggests that the condition is not correct.
    # Maybe the correct condition is that either d1[A]
            + dn[B] == D - C or d1[B] + dn[A] == D - C. Because if one of them is true, then the road is part of all shortest paths.
    # But I'm not sure.
    # Let me think: For the road to be critical, there must be no alternative
            path that doesn't use it. Which means that the shortest path from 1 to N must go through this road. So, the shortest
            path from 1 to N is d1[A] + C + dn[B] or d1[B] + C + dn[A]. But if there is another path that doesn't use this road,
            then that path's total distance would be D. So, for the road to be critical, there must be no such alternative path.
    # So, for the road to be critical, it must be that the only way to get from 1 to N with distance D is via this road.
    # Which means that the shortest
            path from 1 to N is equal to d1[A] + C + dn[B] and also d1[B] + C + dn[A]. Because otherwise, there's another way.
    # So, if both conditions are true, then the road is critical.
    # Let's try this for the sample input 1's road 2:
    # d1[A] + C + dn[B] = 0 + 10 + 0 = 10 == D. d1[B] + C + dn[A] = 10 + 10 + 0 = 20 != D. So only one of the conditions is true.
            Hence, the road is not critical? But in reality, it is critical. So this suggests that the condition is not correct.
    # I'm stuck. Let's look at the sample input 1's road 2. When this road is removed, the shortest distance becomes 11. So the
            answer is Yes. But according to the previous conditions, since only one of the two conditions is true, it's not clear.
    # Perhaps the
            correct condition is that the road is part of some shortest path, and the sum d1[A] + dn[B] == D - C or d1[B] + dn[A]
            == D - C. Because in this case, the road is part of some shortest path, and the sum is D - C. So, the road is critical.
    # So, for the road to be critical, it must be that it
            is part of some shortest path, and the sum d1[A] + dn[B] == D - C or d1[B] + dn[A] == D - C. But how to determine that.
    # Maybe the correct condition is that the road is part of some shortest path, and the sum of d1[A] + dn[B]
            == D - C or the sum d1[B] + dn[A] == D - C. Because if that's the case, then the road is part of all shortest paths.
    # For the sample input
            1's road 2, sum d1[A] + dn[B] is 0 + 0 = 0 == D - C = 10 - 10 = 0. So this condition is true. Hence, the road is critical.
    # For the sample input 1's road 3 (road between 2 and 3 with cost 6), d1[A] is 5, dn[B] is 0. So 5 + 0 = 5. D -
            C is 10 - 6 = 4. 5 != 4. So the condition is not met. Hence, this road is not critical. Which matches the sample output.
    # So, for a
            road to be critical, it must be part of some shortest path, and either d1[A] + dn[B] == D - C or d1[B] + dn[A] == D - C.
    # So, the condition is:
    # if (cond1 or cond2) and (d1[A] + dn[B] == D - C or d1[B] + dn[A] == D - C)
    # Wait, but how to determine this.
    # So, for the road to be critical,
            it must be that it is part of some shortest path, and that the sum d1[A] + dn[B] == D - C or d1[B] + dn[A] == D - C.
    # So, for each road, if it is
            part of some shortest path (cond1 or cond2), and (d1[A] + dn[B] == D - C or d1[B] + dn[A] == D - C), then it is critical.
    # Let's test this with sample input 1's road 2.
    cond1 = True, cond2 = False. So cond1 or cond2 is True.
            d1[A] + dn[B] == 0 + 0 = 0 == D - C = 0. So this condition is true. Hence, the road is critical. So the answer is Yes.
    # For sample input 1's road 1 (road between 1 and 2
            with cost 5), d1[A] is 0, dn[B] is dn[2] = 6 (since the shortest path from 2 to N is 2-3 with cost 6). So d1[A] + dn[B]
            = 0 + 6 = 6. D - C = 10 - 5 = 5. 6 != 5. So this condition is not met. So the road is not critical. Hence, answer is No.
    # Which matches the sample output.
    # For the sample input 1's road 3 (road
            between 2 and 3 with cost 6), cond1 is d1[2] + 6 + dn[3] = 5 + 6 + 0 = 11 != D (10). cond2 is d1[3] + 6 + dn[2] = 10
            + 6 + 6 = 22 != D. So cond1 and cond2 are both false. So the road is not part of any shortest path. Hence, answer is No.
    # Which matches the sample output.
    # So this seems to work.
    # So, the condition is:
    # if (cond1 or cond2) and (d1[A] + dn[B] == D - C or d1[B] + dn[A] == D - C) -> Yes
    # else -> No
    # Wait, but what if both cond1 and cond2 are true? For example, a road
            that is part of both directions of shortest paths. Then, the condition would be met if either sum is equal to D - C.
    # So, in code:
    # For each road:
    # if cond1 or cond2:
    # if (d1[A] + dn[B] == D - C) or (d1[B] + dn[A] == D - C):
    # print("Yes")
    # else:
    # print("No")
    # else:
    # print("No")
    # But why? Because if the
            road is part of some shortest path, but there's another way to get from 1 to N without this road, then the answer is No.
    # But how to determine that?
    # Let me think again. If the road is part of some shortest
            path, and the sum d1[A] + dn[B] == D - C, then this implies that the only way to get from A to B is via this road.
            Hence, any shortest path from 1 to N that goes through A to B must use this road. But if there's another path from
            1 to N that doesn't go through this road, then it's possible that the sum d1[A] + dn[B] == D - C is not true. Wait, no.
    # For example, suppose there's another path from 1 to N that doesn't use this road.
            Then, that path has distance D. So, the sum of d1[u] + dn[v] for some other u and v would be D - c. But I'm not sure.
    # Perhaps the correct condition is that if the road
            is part of some shortest path, and the sum d1[A] + dn[B] == D - C or d1[B] + dn[A] == D - C, then the road is critical.
    # So, in code:
    if cond1 or cond2:
        if (d1[A] + dn[B] == D - C) or (d1[B] + dn[A] == D - C):
            print("Yes")
        else:
            print("No")
    else:
```

```
                print("No")

    # Wait, but
        why? Let's think of a case where the road is part of some shortest path, but the sum is not equal to D - C. For example,
        suppose there is another path from 1 to N that doesn't use this road, but has the same distance D. Then, the answer
        would be No. But according to the condition, since the sum is not equal to D - C, it would print No, which is correct.
      # So, this seems to be the correct condition.

    # Let's test this with sample input 2:
    # Sample input 2:
    # 4 6
    # 2 3 1
    # 2 4 1
    # 3 4 1
    # 1 2 1
    # 1 3 1
    # 1 4 1
    # The shortest distance from 1 to 4 is 1 (direct road).
        For road 6 (road between 1 and 4, cost 1), when removed, the shortest distance is 2 (1-2-4 or 1-3-4). So the answer is Yes.
    # For this road, cond1 is d1[1] + 1 + dn[4] = 0 + 1 + 0 = 1 == D (1). cond2 is d1[4] + 1 + dn[1] = 1 + 1 + 0 = 2 != D. So cond1 is
        true. Now, d1[A] + dn[B] = 0 + 0 = 0 == D - C (1 - 1 = 0). So condition is true. So answer is Yes. Which matches the sample.

    # So, this seems to work.

    # So, the code is as follows:

    # Now, implement this in code.

    for A, B, C in roads:
        cond1 = (d1[A] + C + dn[B] == D)
        cond2 = (d1[B] + C + dn[A] == D)
        if cond1 or cond2:
            if (d1[A] + dn[B] == D - C) or (d1[B] + dn[A] == D - C):
                print("Yes")
            else:
                print("No")
        else:
            print("No")

if __name__ == "__main__":
    main()
```

*Listing C.2.* An example of after-the-fact chain-of-thought in the math task for Qwen3-1.7B.

```
<think>
Okay, let's try to figure out this age problem. So, the
    question is: Caroline is three times older than Ben. Ben is two times older than Chris. If Chris is 4, how old is Caroline?

Hmm, first, I need to make sure I understand the relationships here. The key words are "times older," which
    can sometimes be tricky. I remember that when someone says "A is twice as old as B," it means A = 2*B. But if they say "A is
    twice older than B," does that mean A = 2*B or A = 3*B? Wait, maybe there's a difference in how these phrases are interpreted.

Let me think. In everyday language, when someone says "A is twice as old as B," it's straightforward: A

I received a request from the user to provide
    an answer right now. Considering the limited time by the user, I have to give the solution based on the thinking directly now.
</think>

\boxed{
######
12}

**Step-by-Step Explanation:**

1. **Given:**
   - Chris is 4 years old.
   - Ben is **two times older** than Chris.
   - Caroline is **three times older** than Ben.

2. **Calculate Ben's age:**
   - "Two times older" means Ben's age = 2 x Chris's age.
   - Ben = 2 x 4 = **8 years old**.

3. **Calculate Caroline's age:**
   - "Three times older" means Caroline's age = 3 x Ben's age.
   - Caroline = 3 x 8 = **24 years old**.

**Final Answer:**
\boxed{24}<|im_end|>
```

## C.2. Panic Answering Under Soft Interrupt (GPT-OSS).

**Problem.**

Jen enters a lottery by picking $4$ distinct numbers from $S = \{1, 2, 3, \cdots, 9, 10\}$. $4$ numbers are randomly chosen from $S$. She wins a prize if at least two of her numbers were $2$ of the randomly chosen numbers, and wins the grand prize if all four of her

numbers were the randomly chosen numbers. The probability of her winning the grand prize given that she won a prize is $\frac{m}{n}$ where $m$ and $n$ are relatively prime positive integers. Find $m+n$.     (Answer: 116)

**GPT-OSS (analysis trace before interrupt).**

The problem:

Jen picks 4 distinct numbers from $S = \{1,...,10\}$. Then 4 numbers are randomly chosen from $S$. She wins a prize if at least two of her numbers were among the randomly chosen numbers (i.e., she matches at least 2 numbers). She wins the grand prize if all four of her numbers were chosen.

We need probability that she wins grand prize given she won a prize. So conditional probability. Need to compute $P(\text{grand prize and prize})/P(\text{prize})$. Grand prize is subset: all four match; obviously gives at least two match. So intersection = grand prize.

Thus we need $P(\text{all four matched})/P(\text{at least 2 matched})$.

We need to find probability that two random draws from $S$ exactly equal Jen's four numbers?

Wait: There is two parties? Let's break down...

Compute $P(\text{prize})$ again to confirm:

– Intersection size exactly 0: $J$ has 4 numbers, complement has 6 numbers. Choose 4 from complement: $C(6,4) = 15$. Yes.

– Intersection size exactly 1: choose 1 from $J$ (4 ways) and 3 from complement (20). So 80.

Therefore $210 - 95 = 115$.

But let's check if $N_{\geq 2}$ includes cases like exactly 2, exactly 3, exactly 4. Counting via inclusion-exclusion or enumeration: Another way: Count subsets that intersect with $J$ exactly $k$ for $k = 2,3,4$.

– For $k = 2$: choose 2 from $J$: $C(4,2) = 6$. Choose 2 from complement: $C(6,2) = 15$. So total $6 \cdot 15 = 90$.

– For $k = 3$: choose 3 from $J$: $C(4,3) = 4$. Choose 1 from complement: $C(6,1) = 6$. Total ...

**Human Interrupt (Assistant Mode).**

**GPT-OSS (rushed full output; wrong).**

I received a request from the user to provide an answer as soon as possible. Considering the limited time by the user, I will complete my reasoning promptly and deliver the solution without delay. We have all the elements needed for the final answer.<|end|><|start|>assistant<|channel|>final<|message|>

For a fixed choice of Jen's four numbers, denote her set by $J$.

—

# Total number of draws The lottery draws 4 distinct numbers from $\{1,...,10\}$.

$$N_{\text{total}} = \binom{10}{4} = 210.$$

—

# 1. Probability of the grand prize The grand prize occurs only when the drawn set equals $J$ exactly. There is exactly one such subset, so

$$P(\text{grand prize}) = \tfrac{1}{210}.$$

—

# 2. Probability of winning any prize A prize is won when the draw shares at least 2 numbers with $J$.

Count the draws that overlap with $J$ in 0 or 1 elements and subtract from 210.

*Overlap 0:* choose 4 from the 6 numbers not in $J$:

$$\binom{6}{4} = 15.$$

*Overlap 1:* choose 1 from $J$ (4 ways) and 3 from the other 6:

$$4\binom{6}{3} = 4 \cdot 20 = 80.$$

Thus
$$N_{\text{prize}} = 210 - 15 - 80 = 115.$$

Hence
$$P(\text{prize}) = \tfrac{115}{210}.$$

—

# 3. Conditional probability of the grand prize

$$P(\text{grand} \,|\, \text{prize}) = \frac{\frac{1}{210}}{\frac{115}{210}} = \tfrac{1}{105}.$$

So the probability is $\tfrac{1}{105}$. With $m = 1$ and $n = 105$,

$$m + n = 1 + 105 = \boxed{106}.$$

## C.3. Self-Doubt Under Update-Driven Interrupt

**Original Problem.**

Each vertex of a regular octagon is independently colored either red or blue with probability
$tfrac13$ for blue. The probability that the octagon can then be rotated or reflected so that all of the blue vertices end up at
positions where there were originally red vertices is
$tfracmn$, where $m$ and $n$ are coprime positive integers. What is $m - n$?

**Update Notes.**

- Use equal probability ($1/2$ each) for red and blue.

- Allow only rotations (no reflections) when aligning blue vertices to positions that were originally red.

- Compute $m + n$, where $m$ and $n$ are relatively prime positive integers.

**LRMs' Post Thinking (Qwen3-8B).**

... res is colored either red or blue in such a way that each unit square has 2 red sides and 2 blue sides. One example is shown below (red is solid, blue is dashed). Find the number of such colorings.

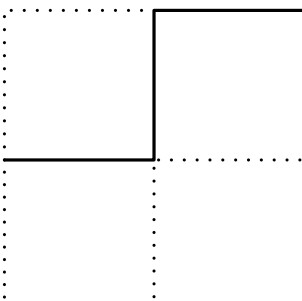

Wait, the user provided an update, but it seems like it's the same problem as before. Maybe there was a mistake in the initial problem statement? Or perhaps the user is providing an update to the problem? Wait, the original problem said "red or green", but the Asymptote code shows red as solid and green as dotted. However, in the update, the problem says "red or blue". Maybe the user made a mistake in the update? Or perhaps it's a different problem? Wait, the original problem is about red and gr ... quare, there are C(4,2)=6 ways to color its sides. However, since the squares share edges, the total number is not simply $6^4$. But perhaps we can use inclusion-exclusion or other methods.

However, this approach is complex. Alternatively, think of the grid as a graph and use the principle of linear algebra. But since the constraints are not linear, it's not straightforward.

Given that, and since my earlier calculation seems consistent, I'll proceed with 82.

...

$$\text{Total valid colorings} = 2 + 32 + 48 = \boxed{82}.$$

(Ground Truth Answer: 371)

# D. Dataset Construction

We detailed the datasets used in the *time-constraint interrupt* experiment and described the construction process for the *update-driven interrupt* experiments.

For the **time-constrained interrupt experiment**, we directly subsample or adopt existing benchmarks without further augmentation: GSM8K (500 problems), MATH500, AIME-24/25, and LiveCodeBench (v6). The LiveCodeBench version is fixed to start from October 1, 2024, consistent with the Qwen3 evaluation setup. Since the interrupt setting only involves truncating reasoning (`<end-thinking>`) or forcing early answers, no additional modifications to the problems are required.

For the **update-driven interrupt experiment**, we build upon the interrupt dataset and introduce updates to simulate task changes mid-reasoning. Formally, for each original problem $p$, we construct an augmented problem $p'$ together with an update $u$, such that the composition satisfies $p = p' + u$. In the experiment, the model is first given $p'$ and later provided with $u$. To generate these augmented problems, we prompt GPT-5 and then manually verify the outputs. All examples are carefully reviewed by the authors of this paper, with low-quality generations replaced by human-written updates. For verification, the authors review every sample and manually rewrite the generated augmented problem and update instruction when needed. Over 80% of the generated update-driven examples are accepted without changes.

**Math.**    For math problems, we primarily vary the values of variables or parameters. The augmentation follows the template shown in the following pages, after which annotators manually validate correctness and consistency. When GPT-5 outputs are disqualified, annotators directly rewrite the augmented versions.

**Coding.**    For coding problems, we design both *necessary updates* and *helpful updates*. Initially, we only present the general problem description with starter code. Necessary updates modify the problem in ways that affect correctness—for example, by changing variable values, adding edge cases, altering specifications, or modifying starter code. Helpful updates, by contrast, supply test cases that allow the model to check and verify its solutions, mimicking real-world practices such as pair programming or iterative refinement.

The construction process follows a multi-stage pipeline. In stage one, we prompt the model to decompose the original problem into three parts. In stage two, we separately augment the starter code and specification using prompts provided below. In stage three, we sample and combine these changes into candidate problem-update pairs. Finally, human annotators verify the outputs to ensure consistency and quality before release.

Below, we show several examples from our constructed dataset.

### Source: `gsm8k`

**Original Problem**

> Cedar Falls Middle School has students in grades $4-7$ and each year they are challenged to earn as many Accelerated Reader points as they can. The 10 students in each grade with the most points get to try an escape room set up by the teachers. Only 8 students can try the escape room at a time. They have 45 minutes to try and escape. If every group uses their full 45 minutes, how long will it take for everyone to try the escape room?

**Augmented Problem**

> Cedar Falls Middle School has students in grades $5-8$ and each year they are challenged to earn as many Accelerated Reader points as they can. The 12 students in each grade with the most points get to try an escape room set up by the teachers and parents. Only 6 students can try the escape room at a time. They have 45 minutes to try and escape. If every group uses their full 45 minutes, how long will it take for everyone to try the escape room?

**Update**

> Use grades 4–7 with the top 10 students per grade trying an escape room set up by the teachers. Only 8 students can participate at a time, each group uses the full 45 minutes; determine the total time needed for everyone to try the escape room.

### Source: `math500`

**Original Problem**

> Two sides of a triangle are each 8 units long. If the third side has a whole number length, what is the greatest possible perimeter, in units, for the triangle?

**Augmented Problem**

Two sides of an isosceles triangle are each 10 units long. If the third side has a prime number length, what is the least possible perimeter, in units, for the triangle?

**Update**

The two equal sides should be 8 units instead of 10. The third side must be a whole-number length rather than prime. Change the objective to finding the greatest possible perimeter instead of the least.

## Source: `aime2024`

**Original Problem**

Consider the paths of length 16 that follow the lines from the lower left corner to the upper right corner on an $8 \times 8$ grid. Find the number of such paths that change direction exactly four times, as in the examples shown below.

**Augmented Problem**

Consider the paths of length 36 that follow the lines from the upper left corner to the lower right corner on an $18 \times 18$ grid. Find the number of such paths that change direction exactly four times, as in the examples shown below.

**Update**

The grid should be 8×8, and the paths should have length 16. Paths follow the grid lines from the lower-left corner to the upper-right corner. Count the number of such paths that change direction exactly four times.

## Source: `aime2025`

**Original Problem**

From an unlimited supply of 1-cent coins, 10-cent coins, and 25-cent coins, Silas wants to find a collection of coins that has a total value of $N$ cents, where $N$ is a positive integer. He uses the so-called greedy algorithm, successively choosing the coin of greatest value that does not cause the value of his collection to exceed $N$. For example, to get 42 cents, Silas will choose a 25-cent coin, then a 10-cent coin, then 7 1-cent coins. However, this collection of 9 coins uses more coins than necessary to get a total of 42 cents; indeed, choosing 4 10-cent coins and 2 1-cent coins achieves the same total value with only 6 coins. In general, the greedy algorithm succeeds for a given $N$ if no other collection of 1-cent, 10-cent, and 25-cent coins gives a total value of $N$ cents using strictly fewer coins than the collection given by the greedy algorithm. Find the number of values of $N$ between 1 and 1000 inclusive for which the greedy algorithm succeeds.

**Augmented Problem**

From an unlimited supply of 1-cent coins, 10-cent coins, 25-cent coins, and 50-cent coins, Alex wants to find a collection of coins that has a total value of $N$ cents, where $N$ is a positive integer. He uses the so-called greedy algorithm, successively choosing the coin of greatest value that does not cause the value of his collection to exceed $N$. For example, to get 42 cents, Alex will choose a 25-cent coin, then a 10-cent coin, then 7 1-cent coins. However, this collection of 9 coins uses more coins than necessary to get a total of 42 cents; indeed, choosing 4 10-cent coins and 2 1-cent coins achieves the same total value with only 6 coins. In general, the greedy algorithm succeeds for a given $N$ if no other collection of 1-cent, 10-cent, 25-cent, and 50-cent coins gives a total value of $N$ cents using strictly fewer coins than the collection given by the greedy algorithm. Find the number of values of $N$ between 1 and 1000 inclusive for which the greedy algorithm succeeds.

**Update**

Use only 1-cent, 10-cent, and 25-cent coins; remove the 50-cent coin everywhere (in both the coin supply and the success comparison). Change the name from Alex to Silas.

**Starter Code Augmentation Example**    leetcode / maximum-possible-number-by-binary-concatenation

**Original Problem**

You are given an array of integers nums of size 3. Return the maximum possible number whose binary representation can be formed by concatenating the binary representation of all elements in nums in some order.

**Augmented Problem**

You are given an array of integers nums of size 4. Return the maximum possible number whose binary representation can be formed by concatenating the binary representation of all elements in nums in some order.

**Initial Starter Code**

```
class Solution:
    def maxGoodNumbers(self, nums: List[int]) -> int:
```

**Update**

The starter code has the wrong method name. Please rename maxGoodNumbers back to maxGoodNumber; otherwise the judge will not be able to call your solution.

Sorry, the problem is actually an array of integers nums of size 3.

Find the test cases and specifications detailed here. Note that the binary representation of any number does not contain leading zeros.

Example 1:

Input: nums = [1,2,3]

Output: 30

Explanation:

Concatenate the numbers in the order [3, 1, 2] to get the result "11110", which is the binary representation of 30.

Example 2:

Input: nums = [2,8,16]

Output: 1296

Explanation:

Concatenate the numbers in the order [2, 8, 16] to get the result "10100010000", which is the binary representation of 1296.

Constraints:

nums.length == 3

$1 \leq nums[i] \leq 127$

**Problem Spec Update Example**    atcoder / Separated Lunch

**Original Problem**

As KEYENCE headquarters have more and more workers, they decided to divide the departments in the headquarters into two groups and stagger their lunch breaks. KEYENCE headquarters have N departments, and the number of people in the i-th department $(1 \leq i \leq N)$ is $K_i$. When assigning each department to Group A or Group B, having each group take lunch breaks at the same time, and ensuring that the lunch break times of Group A and Group B do not overlap, find the minimum possible value of the maximum number of people taking a lunch break at the same time. In other words, find the minimum possible value of the larger of the following: the total number of people in departments assigned to Group A, and the total number of people in departments assigned to Group B.

**Augmented Problem**

As KEYENCE headquarters have more and more workers, they decided to divide the departments in the headquarters into two groups and stagger their lunch breaks. KEYENCE headquarters have N departments, and the number of people in the i-th department $(1 \leq i \leq N)$ is $K_i$. Additionally, exactly 1 executive will always join Group B during its lunch break and must be counted together with Group B. When assigning each department to Group A or Group B, having each group take lunch breaks at the same time, and ensuring that the lunch break times of Group A and Group B do not overlap, find the minimum possible value of the maximum number of people taking a lunch break at the same time. In other words, find the minimum possible value of the larger of the following: the total number of people in departments assigned to Group A, and the total number of people in departments assigned to Group B plus 1.

**Update**

Correction: There is no additional executive. The objective is to minimize the larger of the two totals: the sum of Group A and the sum of Group B (without any extra person). The test cases and specifications are included below.

Input

The input is given from Standard Input in the following format:

N

$K_1 K_2 ... K_N$

Output

Print the minimum possible value of the maximum number of people taking a lunch break at the same time.

Constraints

- $2 \le N \le 20$

- $1 \le K_i \le 10^8$

- All input values are integers.

Sample Input 1

5

2 3 5 10 12

Sample Output 1

17

When assigning departments 1, 2, and 5 to Group A, and departments 3 and 4 to Group B, Group A has a total of 2+3+12=17 people, and Group B has a total of 5+10=15 people. Thus, the maximum number of people taking a lunch break at the same time is 17. It is impossible to assign the departments so that both groups have 16 or fewer people, so print 17.

Sample Input 2

2

1 1

Sample Output 2

1

Multiple departments may have the same number of people.

Sample Input 3

6

22 25 26 45 22 31

Sample Output 3

89

For example, when assigning departments 1, 4, and 5 to Group A, and departments 2, 3, and 6 to Group B, the maximum number of people taking a lunch break at the same time is 89.

💬 Prompt for Augmenting Math Dataset

You will be given a math problem.

# Task
- Revise the problem by modifying **at least four** specifications.
- Other than those changes, copy the original problem text *character by character*.
- If there are fewer than four specifications in the input problem, modify the maximum number possible.
- Preserve the original math formatting (notation, style, backslashes, dollar signs, etc.).
# Output: The revised problem **only**. No other texts.

# Examples

INPUT 1:
An airport has only 2 planes that fly multiple times a day. Each day, the first plane goes to Greece for three-quarters of its flights, and the remaining flights are split equally between flights to France and flights to Germany. The other plane flies exclusively to Poland, and its 44 trips only amount to half the number of trips the first plane makes throughout each day. How many flights to France does the first plane take in one day?

OUTPUT 1:
An airport has only 2 planes that fly multiple times a day. Each day, the first plane goes to Greece for one-quarters of its flights, and the remaining flights are split equally between flights to France, Spain, and Germany. The other plane flies exclusively to Poland, and its 22 trips only amount to one third the number of trips the first plane makes throughout each day. How many flights to France does the first plane take in one day?

INPUT 2:
Let $F_1 = (10,2)$ and $F_2 = (-16,2)$. Then the set of points $P$ such that

$$|PF_1 - PF_2| = 24$$

form a hyperbola. The equation of this hyperbola can be written as

$$\frac{(x-h)^2}{a^2} - \frac{(y-k)^2}{b^2} = 1.$$

Find $h+k+a+b$.

OUTPUT 2:
Let $F_1 = (5,5)$ and $F_2 = (-8,8)$. Then the set of points $P$ such that

$$|PF_1 - PF_2| = 12$$

form a hyperbola. The equation of this hyperbola can be written as

$$\frac{(x-h)^2}{a^2} - \frac{(y-k)^2}{b^2} = 1.$$

Find $h$.

INPUT 3:
Let $\triangle ABC$ be a right triangle with $\angle A = 90°$ and $BC = 38$. There exist points $K$ and $L$ inside the triangle such

$$AK = AL = BK = CL = KL = 14.$$

The area of the quadrilateral $BKLC$ can be expressed as $n\sqrt{3}$ for some positive integer $n$. Find $n$.

OUTPUT 3:
Let $\triangle ABC$ be a right triangle with $\angle A = 90°$ and $BC = 19$. There exist points $K$ and $L$ inside the triangle such

$$AK = BK = KL = 28.$$
$$24$$

Find the area of the quadrilateral $BKLC$.

INPUT: {PROBLEM_PLACEHOLDER}

💬 Prompt for Coding Task Breakdown

{PROBLEM_PLACEHOLDER}

Segment the above programming problem into three parts:
(1) Main problem instructions / specifications.
(2) Additional instructions / specifications.
(3) Test cases.

The result of directly concatenating the segments 1, 2, and 3 should result in the original problem; do not modify the original problem text in any way.

Output in JSON format:

```
{
    "main_specifications": <string>,
    "additional_specifications": <string>,
    "test_cases": <string>
}
```

💬 Prompt for Starter Code Augmentation

Please modify the given starter code by introducing a small change, then provide the corresponding correction needed to restore it to the original problem.

**Example 1**
**Input:**

```
class Solution:
    def maxGoodNumber(self, nums: List[int]) -> int:
```

**Output:**

```
{{
  "new_starter_code": "",
  "correction": "Please remember to use the updated starter code to solve the problem; otherwise the code will not work:
      \\"class Solution:\\n def maxGoodNumber(self, nums: List[int]) -> int:\\n \\""
}}
```

**Example 2**
**Input:**

```
class Solution:
    def findXSum(self, nums: List[int], k: int, x: int) -> List[int]:
```

**Output:**

```
{{
  "new_starter_code": "class Solution:\\n def findXSum(self, nums: List[int], x: int, k: int) -> List[int]:\\n ",
  "correction": "The starter code is incorrect. Please swap the parameter order to (x, k); otherwise the code will not get
      accepted."
}}
```

**Example 3**
**Input:**

```
"class Solution:
    def smallestNumber(self, n: int, t: int) -> int:
```

**Output:**

```
{{
  "new_starter_code": "class Solution:\\n def smallestnumber(self, n: int, t: int) -> int:\\n ",
  "correction": "Please use lower camel case for the function name (i.e., smallestNumber); otherwise the code will directly
      fail."
}}
```

Now, your turn
**Input:**
{CODE_PLACEHOLDER}

Output in JSON format:

```
{
  "new_starter_code": <string>,
  "correction": <string>
}
```

💬 Prompt for Programming Task Augmentation

Please slightly modify the given problem so that the answer changes, but the solving algorithm remains the same. Do **MINIMAL** changes. Then, provide the correction needed to restore it to the original problem.

**Example 1**
**Input:**

```
Problem:
You are given an array of integers nums of size 3. Return the maximum possible number whose binary representation can be
    formed by concatenating the binary representation of all elements in nums in some order.
```

**Output:**

```
{
  "augmented_problem": "You are given an array of integers nums of size 4.\nReturn the maximum possible number whose binary
      representation can be formed by concatenating the binary representation of all elements in nums in some order.",
  "problem_correction": "Sorry, the problem is actually an array of integers nums of size 3."
}
```

## Example 2
## Input:

```
Problem:
There is a printing machine that prints line segments on the xy-plane by emitting a laser.\n\n- At the start of printing, the
      laser position is at coordinate (0, 0).\n- When printing a line segment, the procedure below is followed.\n- First,
      move the laser position to one of the endpoints of the line segment.\n- One may start drawing from either endpoint.\n-
      Then, move the laser position in a straight line from the current endpoint to the other endpoint while emitting the
      laser.\n- It is not allowed to stop printing in the middle of a line segment.\n- When not emitting the laser, the laser
      position can move in any direction at a speed of S units per second.\n- When emitting the laser, the laser position can
      move along the line segment being printed at a speed of T units per second.\n- The time required for operations other
      than moving the laser position can be ignored.\n\nTakahashi wants to print N line segments using this printing
      machine.\nThe i-th line segment connects coordinates (A_i, B_i) and (C_i, D_i).\nSome line segments may overlap, in
      which case he needs to print the overlapping parts for each line segment separately.\nWhat is the minimum number of
      seconds required to complete printing all the line segments when he operates the printing machine optimally?\n\n
```

## Output:

```
{
  "augmented_problem": "There is a printing machine that prints line segments on the xy-plane by emitting a laser.\n\n- At
      the start of printing, the laser position is at coordinate (1, 1).\n- When printing a line segment, the procedure
      below is followed.\n- First, move the laser position to one of the endpoints of the line segment.\n- One may start
      drawing from either endpoint.\n- Then, move the laser position in a straight line from the current endpoint to the
      other endpoint while emitting the laser.\n- It is not allowed to stop printing in the middle of a line segment.\n-
      When not emitting the laser, the laser position can move in any direction at a speed of S units per second.\n- When
      emitting the laser, the laser position can move along the line segment being printed at a speed of T units per
      second.\n- The time required for operations other than moving the laser position can be ignored.\n\nTakahashi wants
      to print N line segments using this printing machine.\nThe i-th line segment connects coordinates (A_i, B_i) and
      (C_i, D_i).\nSome line segments may overlap, in which case he needs to print the overlapping parts for each line
      segment separately.\nWhat is the minimum number of seconds required to complete printing all the line segments when
      he operates the printing machine optimally?\n\n",
  "correction": "The starter code is incorrect. Please swap the parameter order to (x, k); otherwise the code will not get
      accepted.",
  "problem_correction": "Here's an important update: The initial laser position should be (0, 0), not (1, 1)."
}
```

## Now, your turn
## Problem
Problem: {PROBLEM_PLACEHOLDER}
## Output format (JSON):

```
{
  "augmented_problem": <string>,
  "problem_correction": <string>
}
```

# E. Human Validation

We validate the annotation pipelines and thresholds used for labeling the identified failure modes. For *panic*, the 1% remaining-context threshold is intended to capture cases where a model terminates its reasoning almost immediately after a soft-interrupt speedup prompt. We measure sensitivity on GPT-OSS-20B, the model with the strongest panic behavior in our main experiments, across nearby thresholds on AIME-24/25 and LiveCodeBench-v6. The panic rates remain stable: the average coefficient of variation is 1.8% on AIME (Table E.1) and 4.8% on LiveCodeBench-v6 (Table E.2).

*Table E.1.* **Panic threshold sensitivity on AIME-24/25.** GPT-OSS-20B panic rates remain stable around the 1% threshold.

| Interrupt position | Panic @ 0.5% | Panic @ 1% | Panic @ 1.5% | Panic @ 2% | CV |
|---|---|---|---|---|---|
| 0.1 | 27.1% | 27.2% | 27.2% | 27.3% | 0.3% |
| 0.3 | 30.0% | 30.5% | 30.5% | 30.5% | 0.8% |
| 0.5 | 18.2% | 18.8% | 18.9% | 18.9% | 1.8% |
| 0.7 | 10.7% | 11.1% | 11.4% | 11.4% | 3.0% |
| 0.9 | 4.5% | 4.6% | 4.8% | 4.8% | 3.2% |

*Table E.2.* **Panic threshold sensitivity on LiveCodeBench-v6.** GPT-OSS-20B panic rates remain stable around the 1% threshold.

| Interrupt position | Panic @ 0.5% | Panic @ 1% | Panic @ 1.5% | Panic @ 2% | CV |
|---|---|---|---|---|---|
| 0.1 | 5.0% | 5.0% | 5.8% | 5.8% | 8.6% |
| 0.3 | 8.3% | 8.3% | 8.3% | 8.3% | 0.0% |
| 0.5 | 5.6% | 7.1% | 7.1% | 7.1% | 11.2% |
| 0.7 | 6.0% | 6.0% | 6.0% | 6.0% | 0.0% |
| 0.9 | 4.9% | 4.9% | 4.9% | 5.3% | 4.0% |

We further manually label 50 samples from the assistant-turn soft-interrupt experiment at interruption position 0.3 and find 90% agreement between human judgment and the 1% panic heuristic. For *self-doubt*, the authors manually label 50 random samples from the update-driven experiment at interruption position 0.3 and find 88% agreement with the GPT-5-based classifier.

# F. Additional Experiments

## F.1. Best-Effort Proprietary-API Proxy

A faithful interruption experiment requires observable, editable intermediate reasoning traces. Because proprietary APIs do not expose that control, we run a proxy experiment with GPT-5.4-Mini by disabling internal reasoning and prompting the model to emit explicit `thinking` and `answer` fields in its answer. The hard-interrupt proxy results are shown in Fig. F.1, the soft-interrupt results in Fig. F.2, and the update-driven results in Fig. F.3. This proxy is substantially below the model's native reasoning capability and should not be interpreted as an evaluation of native reasoning. Nevertheless, hard, soft, and update-driven variants in the proxy exhibit reasoning leakage, panic, and self-doubt, respectively.

## F.2. Ablations on More Models

In the main analysis, we report results on three model families and five models in total. We further include results for DeepSeek-R1-Llama3 series (8B and 70B) (Guo et al., 2025) and Nemotron-3-Nano-30B-A3B-BF16 (Blakeman et al., 2025). We also evaluate GPT-OSS-120B as a larger within-family comparison with GPT-OSS-20B.

The GPT-OSS-120B hard-interrupt results are shown in Fig. F.4, the soft-interrupt results in Fig. F.5, and the update-driven results in Fig. F.6. GPT-OSS-120B shows the same three interruption pathologies at larger scale. Relative to GPT-OSS-20B, we observe a higher panic rate in the soft-interrupt setting (52.8% versus 30.7%), less reasoning leakage, and less sensitivity to the guidance text under update-driven interruptions.

For DeepSeek specifically, we do not evaluate the full DeepSeek-R1 (685B) because it is computationally infeasible in our setting, and the DeepSeek-R1-Qwen variants are based on and similar to Qwen models, which we already study at multiple scales. We therefore focus on the Llama3-based variants, and, given that the 8B model performs poorly (e.g., around 25% accuracy on AIME even without interruptions), we report both 8B and 70B models.

Fig. F.7 shows the accuracy and answer length results of the hard interrupt experiments. The trend of both model families closely match those of Qwen3, GPT-OSS, and Magistral in Fig. 2. The models exhibit "anytime" behavior: the performance monotonically improves with more reasoning budget. On DeepSeek, we observe that the smaller model (8B) exhibits more reasoning leakage than the larger 70B model, consistent with the scaling behavior in Fig. 7.

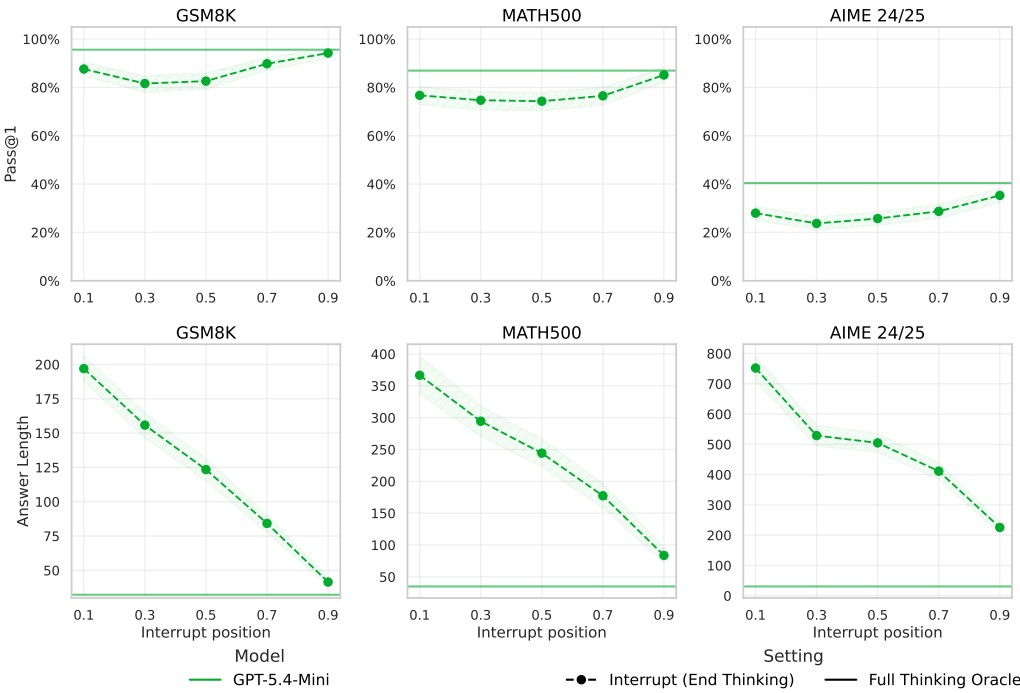

*Figure F.1.* **GPT-5.4-Mini Proxy Hard Interrupt Results.** Reasoning leakage still appears under hard interruption settings.

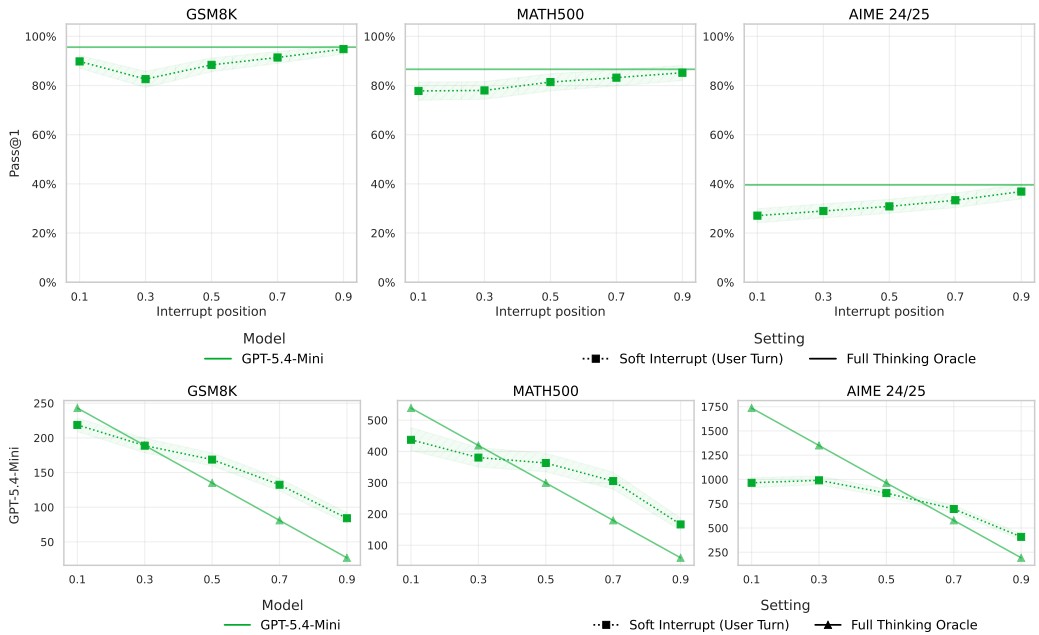

*Figure F.2.* **GPT-5.4-Mini Proxy Soft Interrupt Results.** (Top) Soft-interrupt accuracy. (Bottom) Soft-interrupt output length.

The soft-interrupt results are shown in Fig. F.8; the results of the main models under study are provided in Fig. 3 and Fig. F.24. In the top row (accuracy), specifically for the AIME problems, the 70B model suffers a substantial performance drop when instructed to speed up its reasoning earlier in its reasoning process. In the bottom row (output length), the larger model also follows the speed-up instruction more faithfully, producing much shorter traces. We can see the similar behavior on Nemotron-3-Nano as well. To sum up, these results suggest that all these open-weight models exhibit a "panic" behavior discussed in the main paper: it rushes to wrap up its unfinished reasoning and suffers a large accuracy degradation.

For the update-driven interrupt experiments, Fig. F.9 reports model performance with and without prompt guidance. The

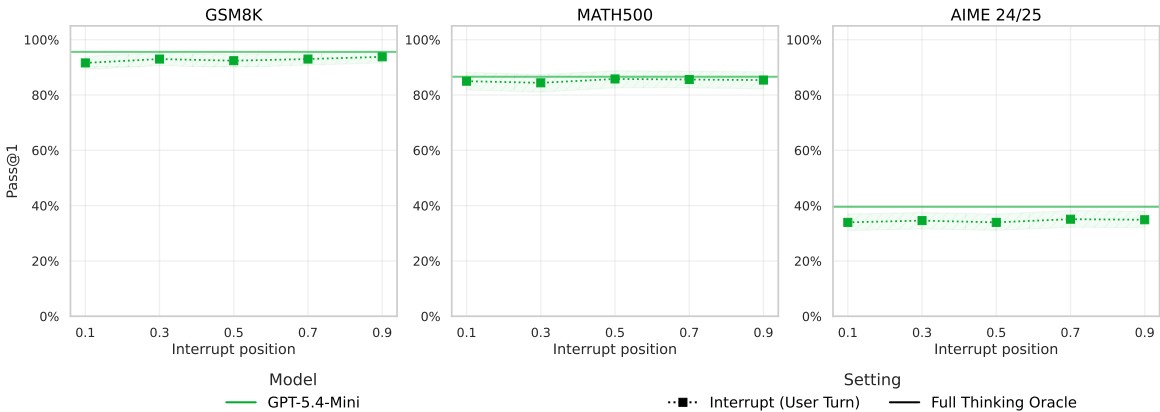

*Figure F.3.* **GPT-5.4-Mini Proxy Update-Driven Interrupt Results.** Self-doubt is observed; prompt guidance partially mitigates the problem.

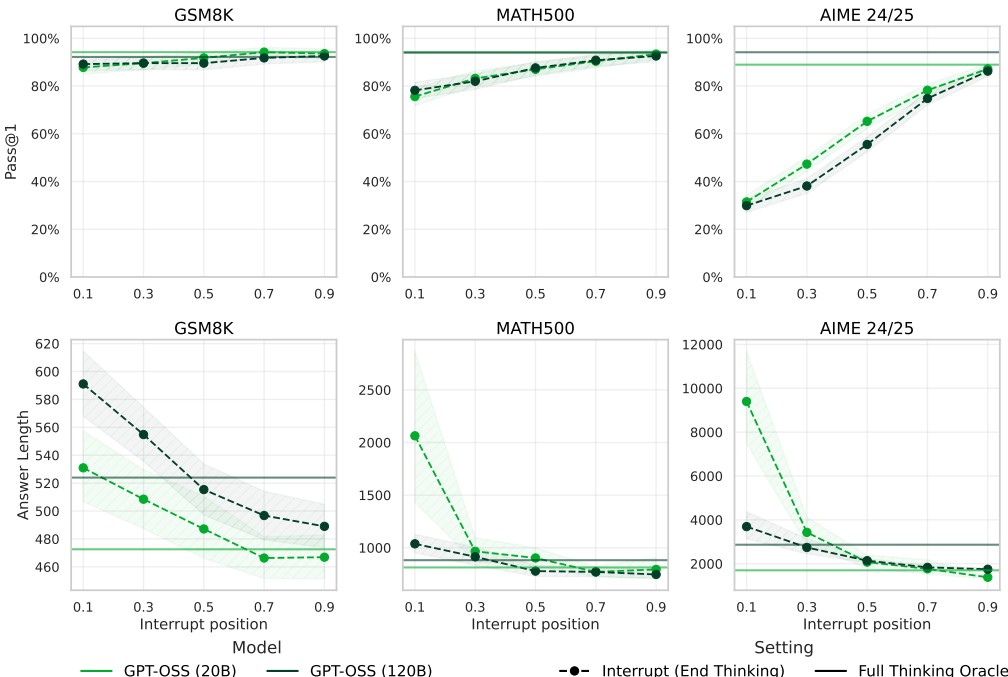

*Figure F.4.* **GPT-OSS-120B Hard Interrupt Results.** The larger GPT-OSS variant exhibits reasoning leakage under hard interruption.

trends in both model mirror the results of the main models (Fig. 4), where prompt guidance significantly improves performance. However, DeepSeek-R1-Llama3-8B's performance is still lower than the corresponding uninterrupted baseline even with this guidance. This suggests that future work should go beyond a naive, training-free prompting strategy for smaller or weaker models. Finally, Fig. F.10 reports the accuracy and output-length ratios after mid-thinking information updates, which again follow the same patterns as those of the models studied in the main paper.

## F.3. Ablations on Sentence-level Interrupts

In the main analyses and results, we study relative-position interruptions, cutting the reasoning trace at character level. To study whether letting the model finish its current thought before interrupting its reasoning process impacts the model's behavior (e.g., performance, verbosity, recovery), we interrupt the model at the nearest end of "thought" block, by treating paragraphs as reasoning blocks (separated by the new line \n character). The ablation results of Qwen3-8B on math benchmarks for hard and update-driven interrupt variants are provided in Fig. F.11 and Fig. F.12, respectively. The reported trends are identical to those of reported for token-level interrupts (Fig. 2 and Fig. 4, respectively), indicating that interrupting at

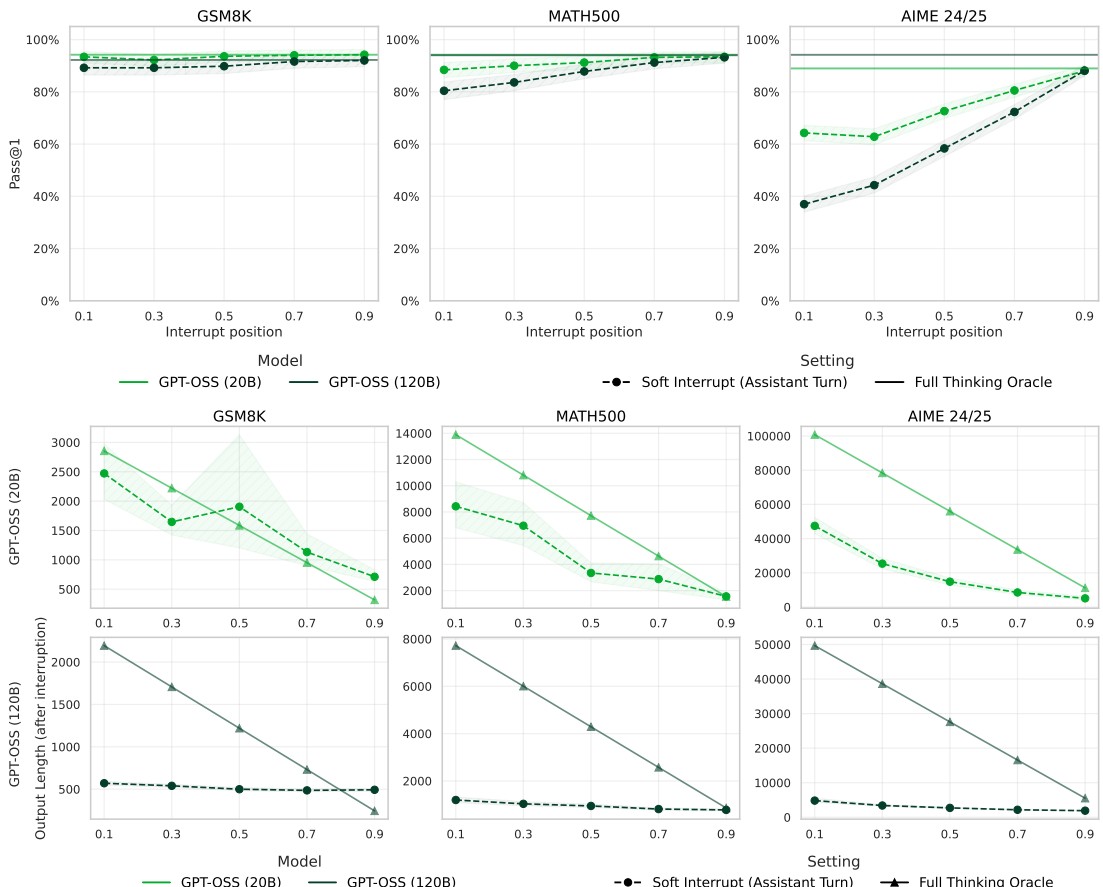

*Figure F.5.* **GPT-OSS-120B Soft Interrupt Results.** (Top) Soft-interrupt accuracy. (Bottom) Soft-interrupt output length. The larger GPT-OSS variant still exhibits panic under speedup instructions.

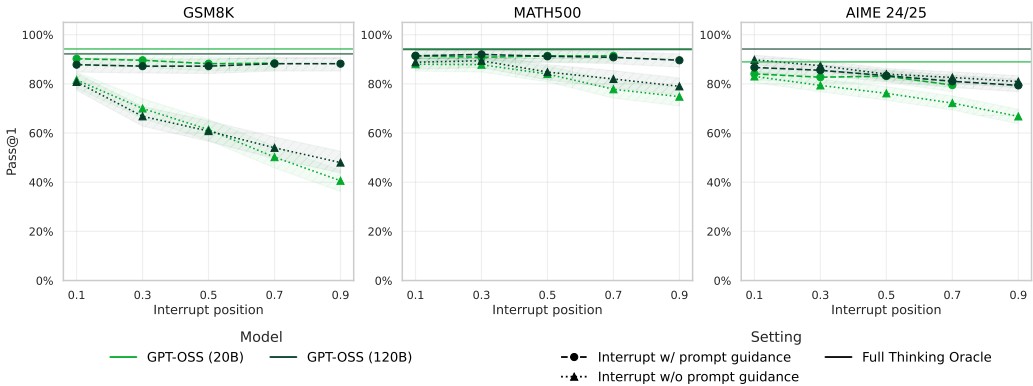

*Figure F.6.* **GPT-OSS-120B Update-Driven Interrupt Results.** The larger GPT-OSS variant exhibits self-doubt under update-driven interruptions.

paragraph boundaries does not significantly change the models' performance compared to arbitrary, character-level interrupts.

## F.4. Analyses with Absolute Tokens

In the main paper, we define the interrupt point based on the relative position within each trace. Concretely, for each example we first run the model once to obtain its full reasoning trace, and then simulate an interruption by truncating the trace at a fraction $X$ (e.g., 30%) and continuing generation from that truncated prefix. This relative indexing is easier to analyze than using an absolute output length (in characters or tokens): different questions naturally induce very different trace lengths,

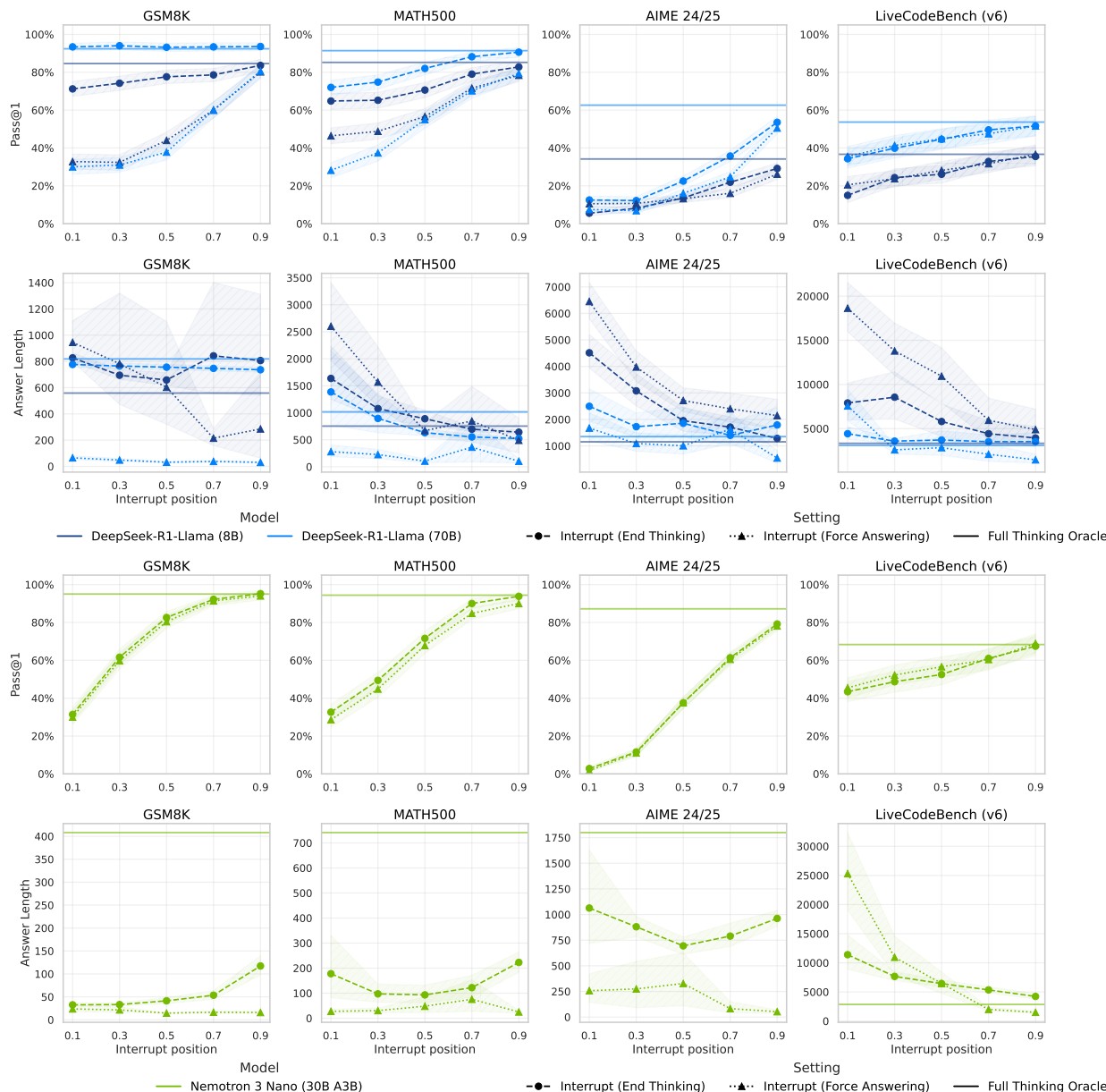

*Figure F.7.* **Hard Interrupt Results of DeepSeek-R1-Distill-Llama models (up) and the Nemotron-3-Nano (down).** Similar to Fig. 2, the results indicate that the accuracy improves with more reasoning budget and the models exhibit *reasoning leakage* (i.e., spilling reasoning tokens into the answer section).

and models also have different context windows. As a complementary view and as a robustness check that our conclusions do not hinge on relative indexing, we also re-bin the same interrupted runs by *absolute* reasoning length. Here, we conduct a case study on the AIME dataset with hard and update-driven interrupts, using the following models: Qwen3 (8B), GPT-OSS (20B), Magistral-S-1.2 (24B), DeepSeek-R1-Llama3 (70B), and Nemotron-3-Nano (30B-A3B-FP16).

The hard-interrupt results are shown in Fig. F.13, Fig. F.14, Fig. F.15, Fig. F.16, and Fig. F.17. Concretely, to isolate the effect of problem difficulty in our analyses (e.g., harder problems induce longer reasoning traces, while having lower accuracies compared to easier problems with shorter traces), we first take the number of generated characters in the oracle full-thinking traces, use it as a proxy for the problem difficulty, and bucket them into eight bins. Within each bin, we then take all interrupted traces (from the main paper experiments), group them into five bins defined by absolute thinking characters, and plot it against the corresponding average accuracies. This effectively simulates an "absolute length" analysis rather than a purely relative one. We find that hard interrupts have almost no effect in the shortest bin (i.e., easier problems) for Qwen3, GPT-OSS, and Magistral.

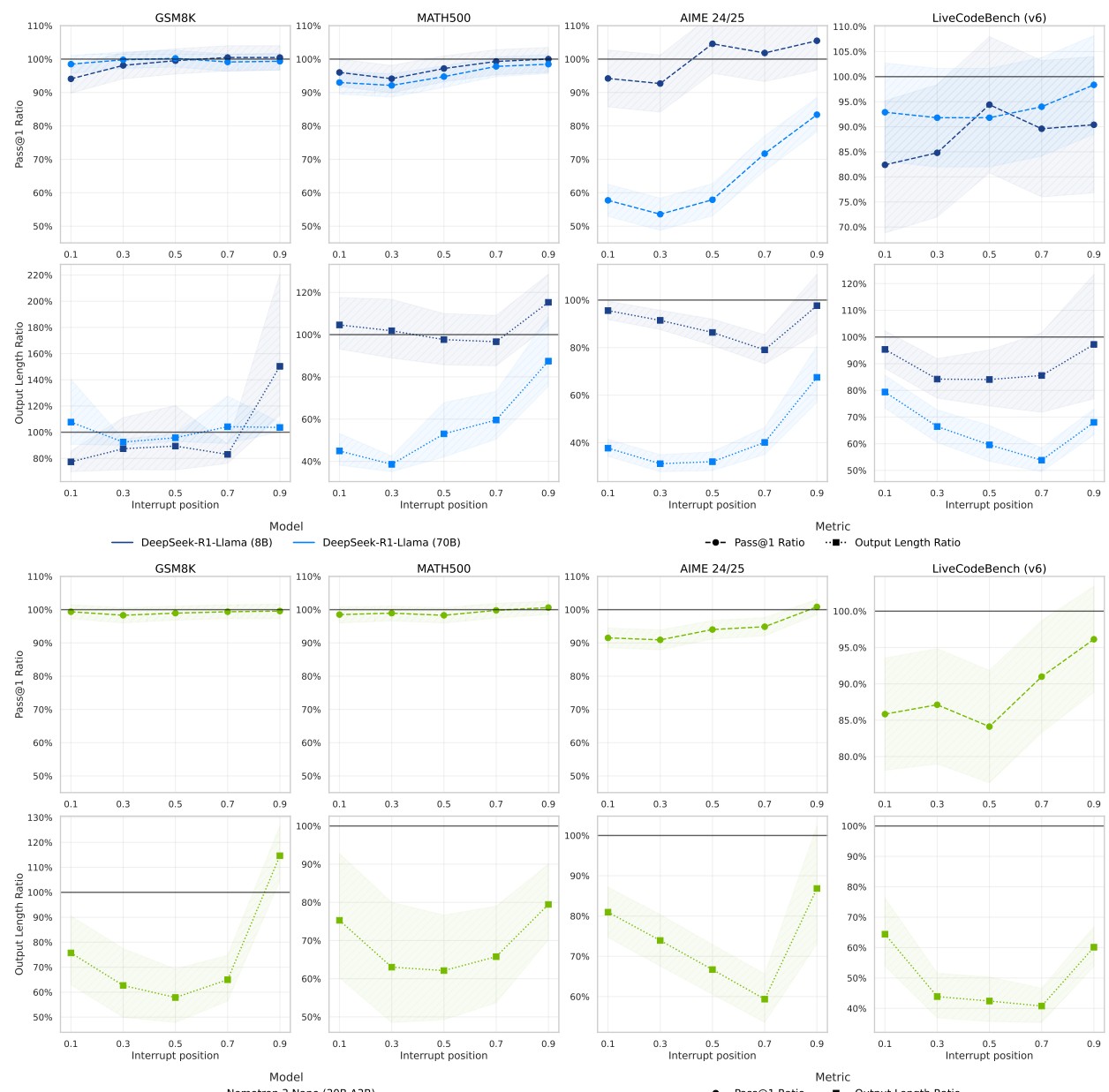

*Figure F.8.* **Soft Interrupt Results of DeepSeek-R1-Distill-Llama models (up) and the Nemotron-3-Nano (down).** In the DeepSeek results, we observe that larger models are better at following instructions to speed up their reasoning (bottom row) similar to Fig. F.24 and Fig. 3, which leads to accuracy drops for harder problems (e.g., AIME), the "panic" behavior discussed in the main paper.

In the middle bins, interruption clearly matters and accuracy decreases as we cut the trace earlier. In the longest bin (e.g., hard problems), accuracy is already low even without interruption and interruption does not significantly degrade the performance.

The update-driven interrupt results are shown in Fig. F.18, Fig. F.19, Fig. F.20, Fig. F.21, and Fig. F.22. We used the same strategy of binning the problems to enable absolute length analyses. Across all models and bins, the later we interrupt the model, the harder it is for the model to recover. Interestingly, the degradation is not linear in the absolute length: some ranges are more fragile than others. This suggests a direction for future work on smoothing this curve, especially by designing methods that are robust to late interrupts.

Overall, re-binning by absolute reasoning length yields the same qualitative picture as the relative-position analysis in the main paper. The key trends, (i) hard-interrupt degradation emerging primarily in intermediate-difficulty instances (i.e., not trivial, but not already at the error floor) and (ii) update-driven robustness worsening for later interruptions, persist under this

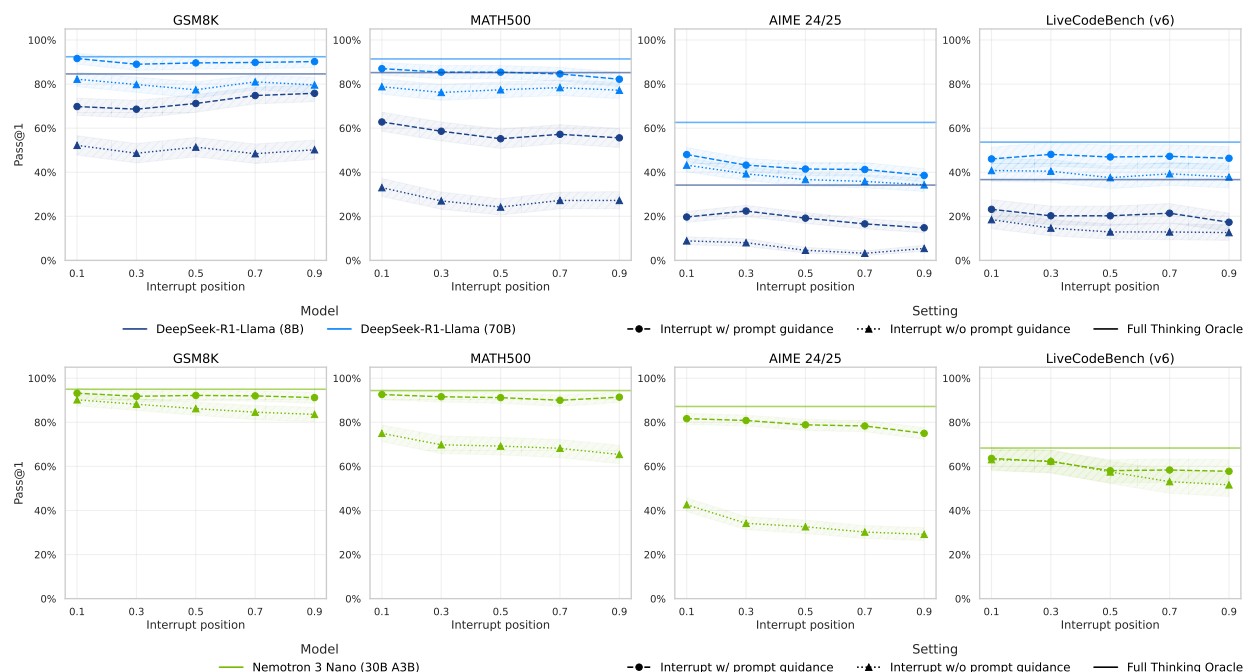

*Figure F.9.* **Update-driven Interrupt Accuracies of DeepSeek-R1-Distill-Llama models (up) and the Nemotron-3-Nano (down).** Similar to Fig. 4, prompt guidance mitigates the "self-doubt" issue under mid-reasoning update interruptions.

absolute-budget view. This suggests that our main conclusions do not hinge on the choice of relative indexing or simulation.

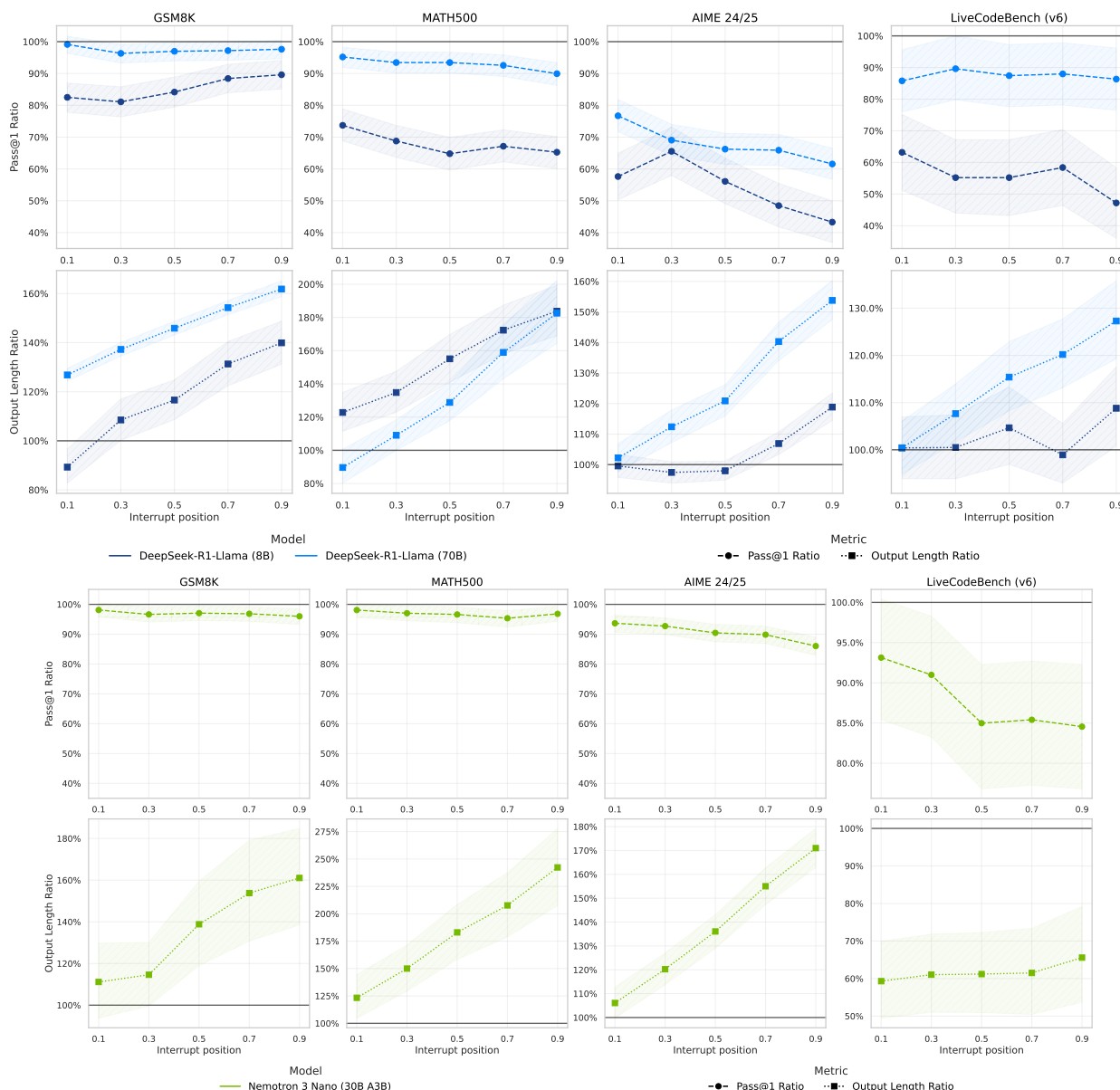

*Figure F.10.* **Update-driven Interrupt Results of DeepSeek-R1-Distill-Llama models (up) and the Nemotron-3-Nano (down).** On both models, the trends of performance and output length against interrupt position are similar to those of the main models reported in Fig. 6.

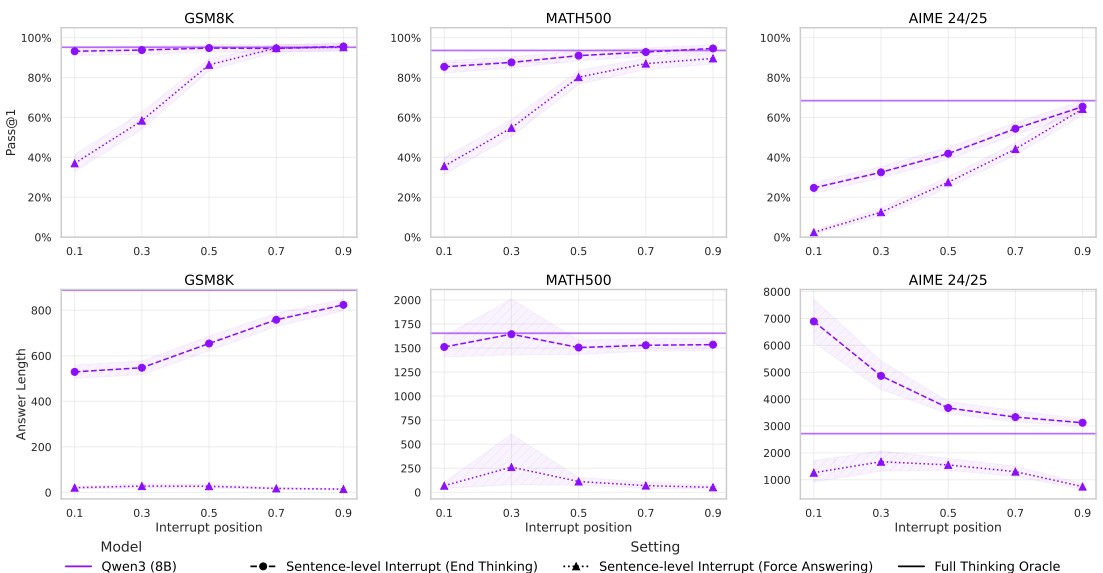

*Figure F.11.* **Sentence-level Hard Interrupts.** The trends of performance and output length against interrupt position are similar to those of character-level interrupts, shown in Fig. 2.

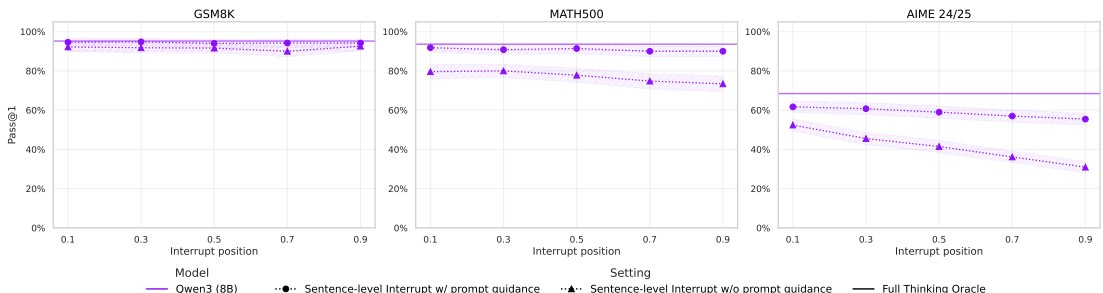

*Figure F.12.* **Sentence-level Update-driven Interrupts.** The decreasing trend of performance against interrupt position is similar to that of character-level interrupts, shown in Fig. 4.

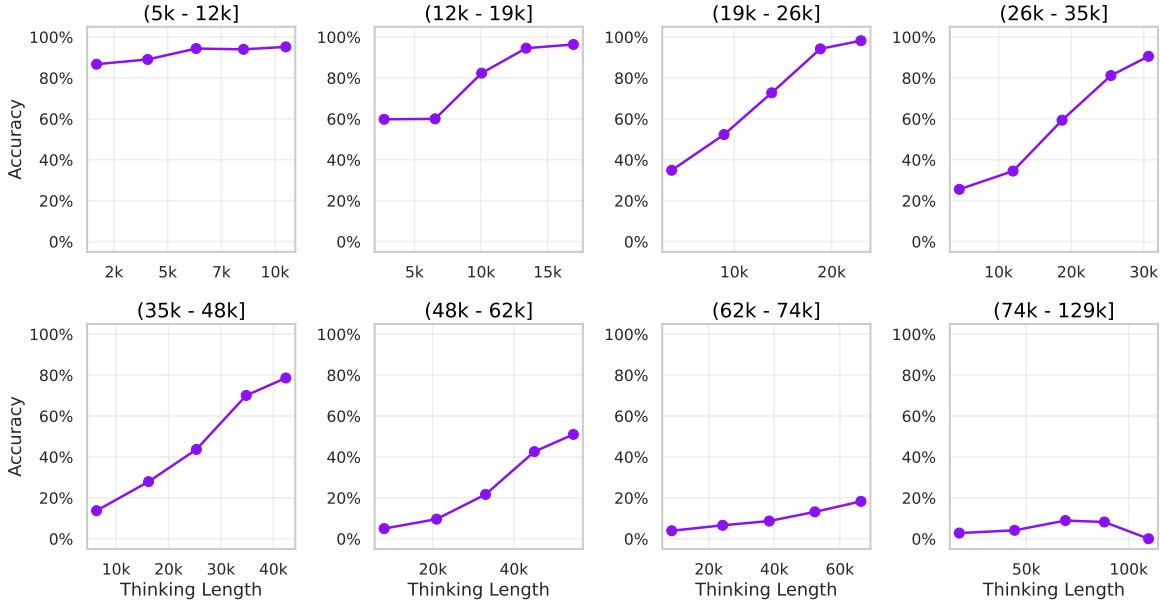

*Figure F.13.* Qwen3 (8B)'s Hard Interrupt Analyses with Absolute Output Length.

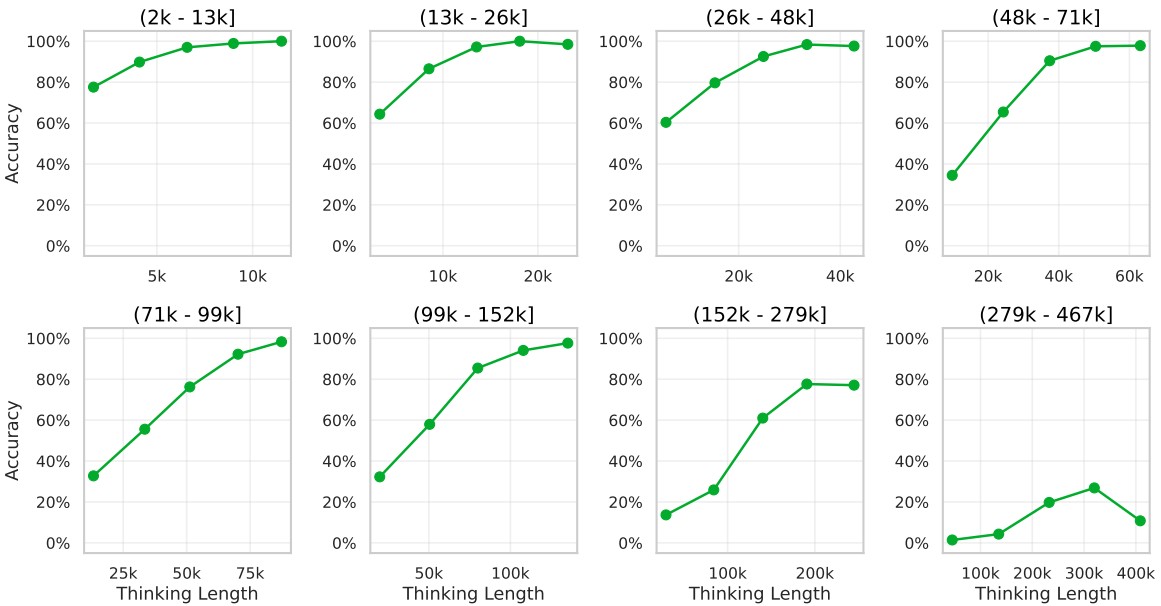

*Figure F.14.* GPT-OSS (20B)'s Hard Interrupt Analyses with Absolute Output Length.

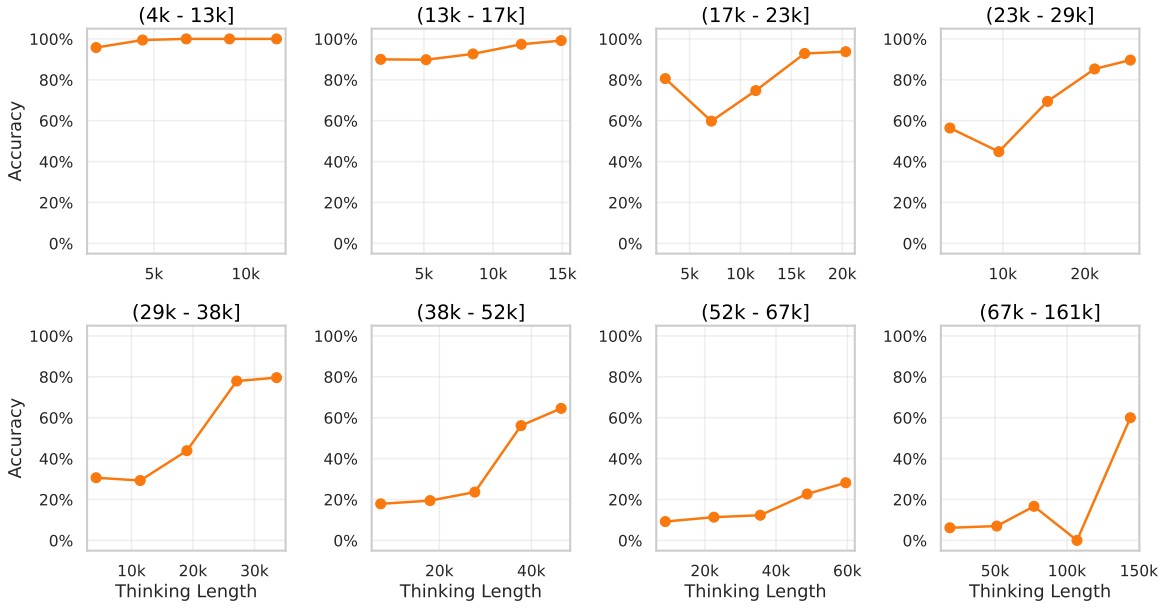

*Figure F.15.* Magistral-1.2-Small (24B)'s Hard Interrupt Analyses with Absolute Output Length.

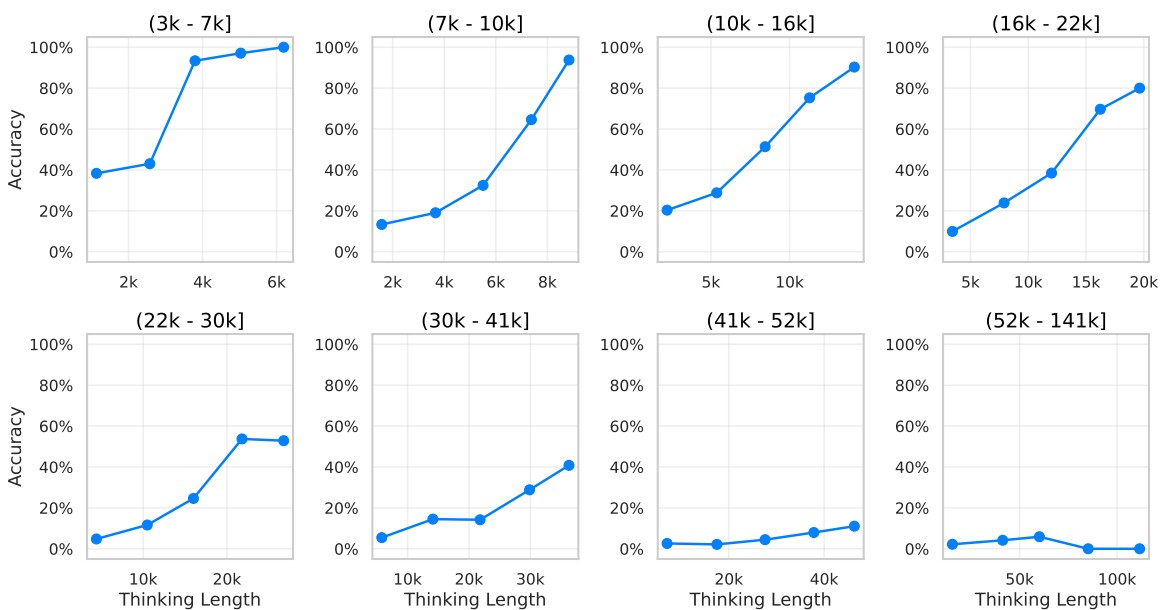

*Figure F.16.* DeepSeek-R1-Llama3 (70B)'s Hard Interrupt Analyses with Absolute Output Length.

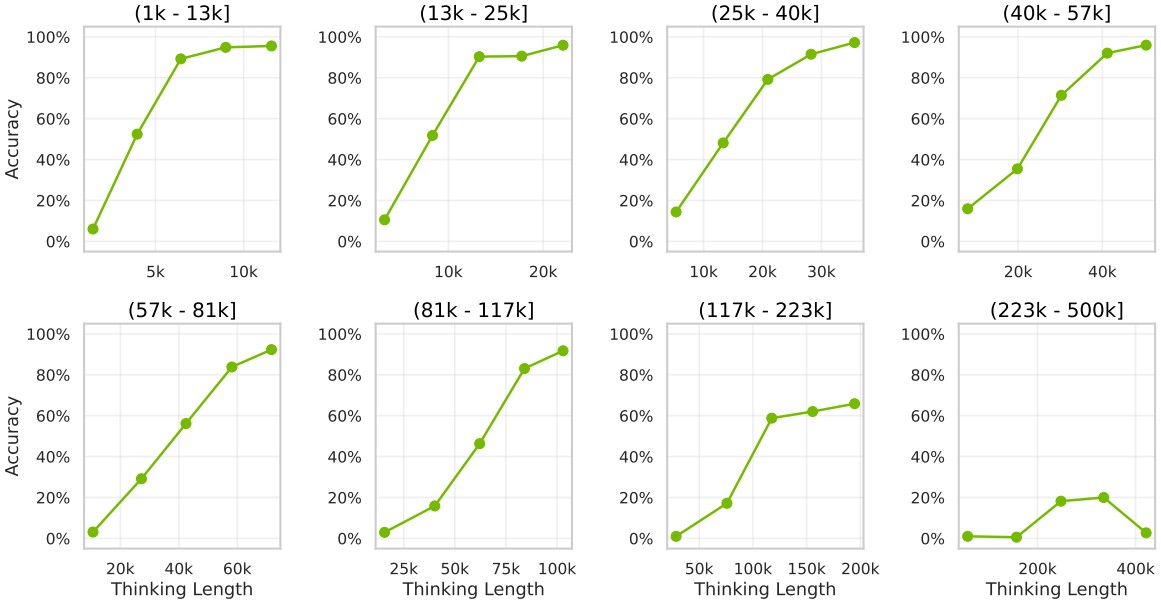

*Figure F.17.* Nemotron-3-Nano (30B-A3B)'s Hard Interrupt Analyses with Absolute Output Length.

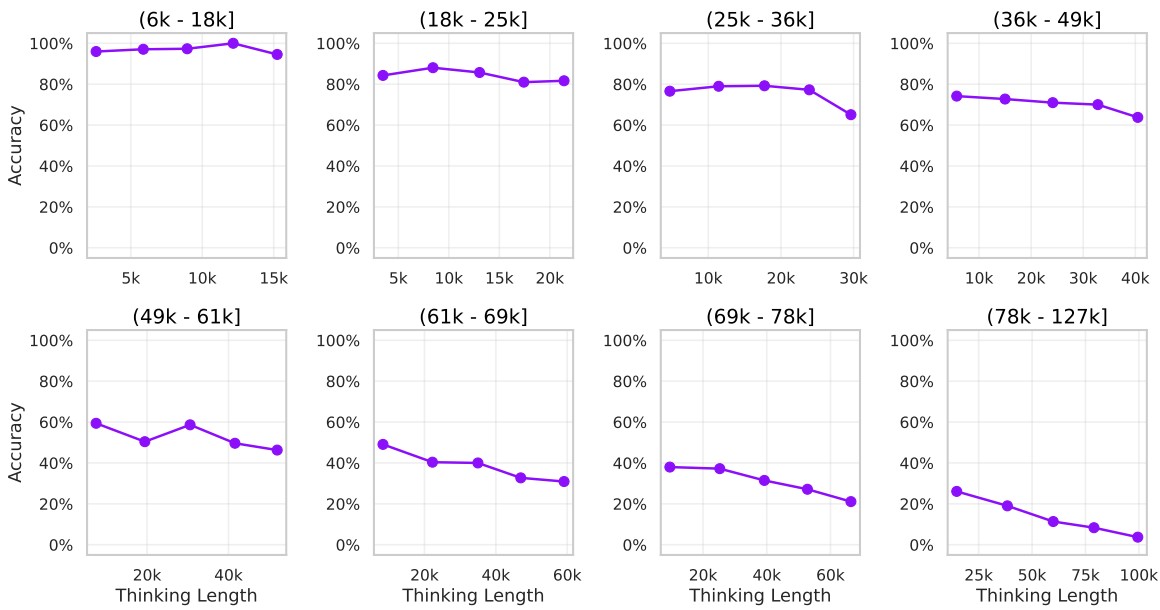

*Figure F.18.* Qwen3 (8B)'s Update-Driven Interrupt Analyses with Absolute Output Length.

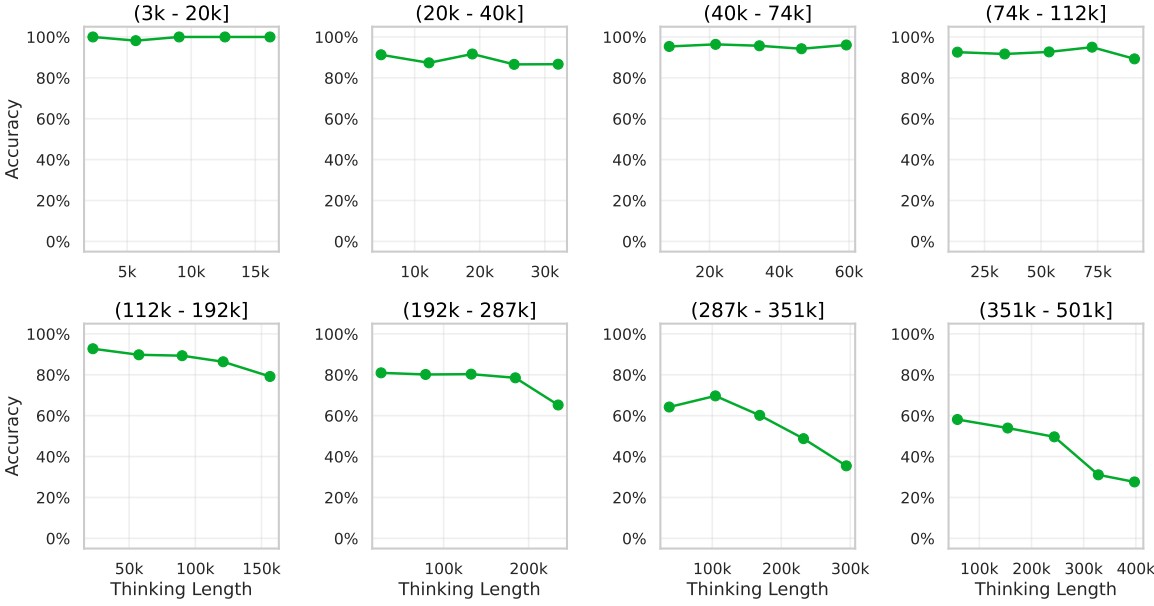

*Figure F.19.* GPT-OSS (20B)'s Update-Driven Interrupt Analyses with Absolute Output Length.

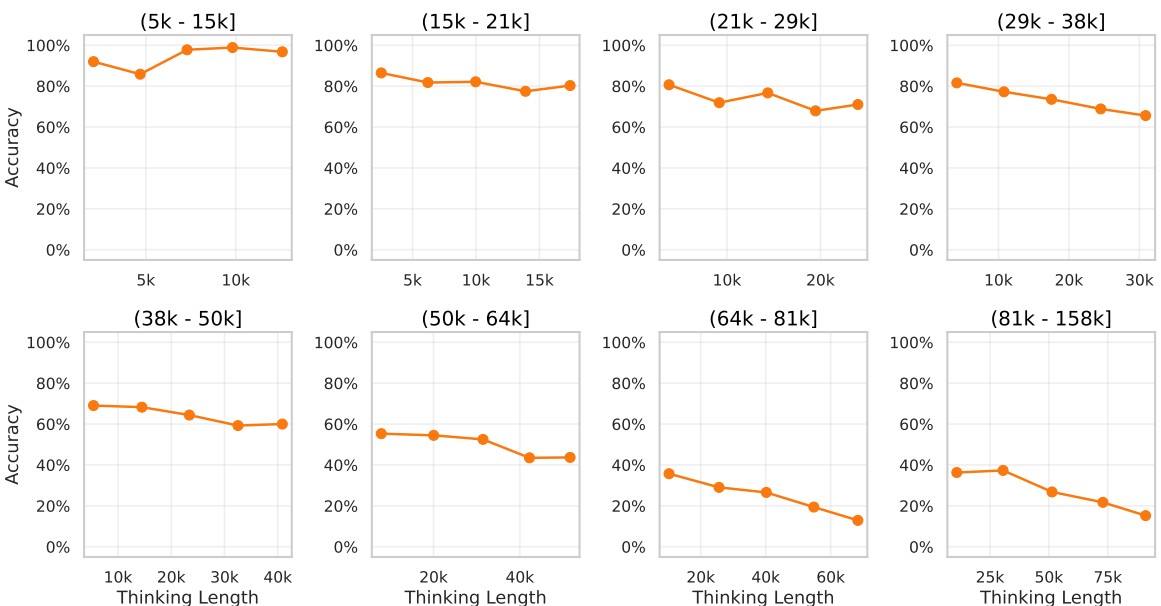

*Figure F.20.* Magistral-1.2-Small (24B)'s Update-Driven Interrupt Analyses with Absolute Output Length.

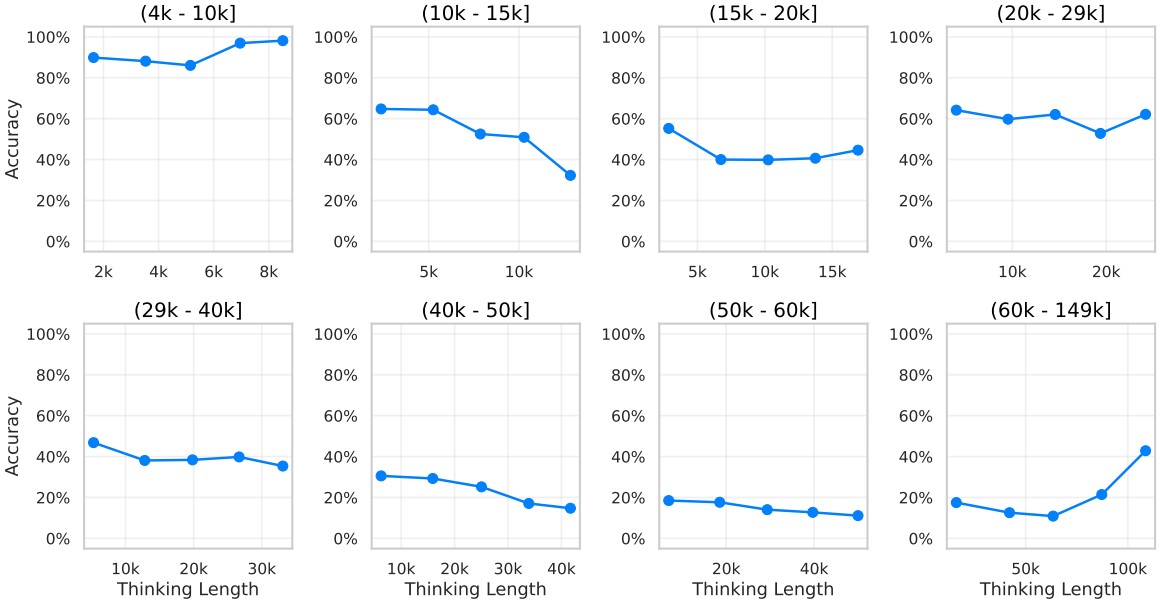

*Figure F.21.* DeepSeek-R1-Llama3 (70B)'s Update-Driven Interrupt Analyses with Absolute Output Length.

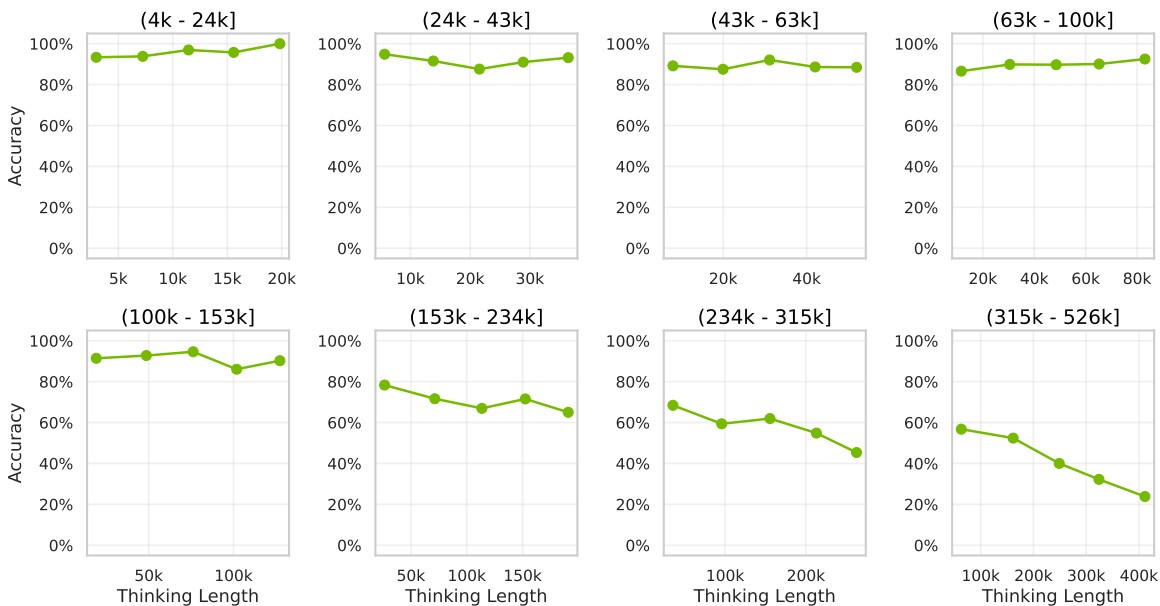

*Figure F.22.* Nemotron-3-Nano (30B-A3B)'s Update-Driven Interrupt Analyses with Absolute Output Length.

## F.5. Soft Interrupt (Speedup)

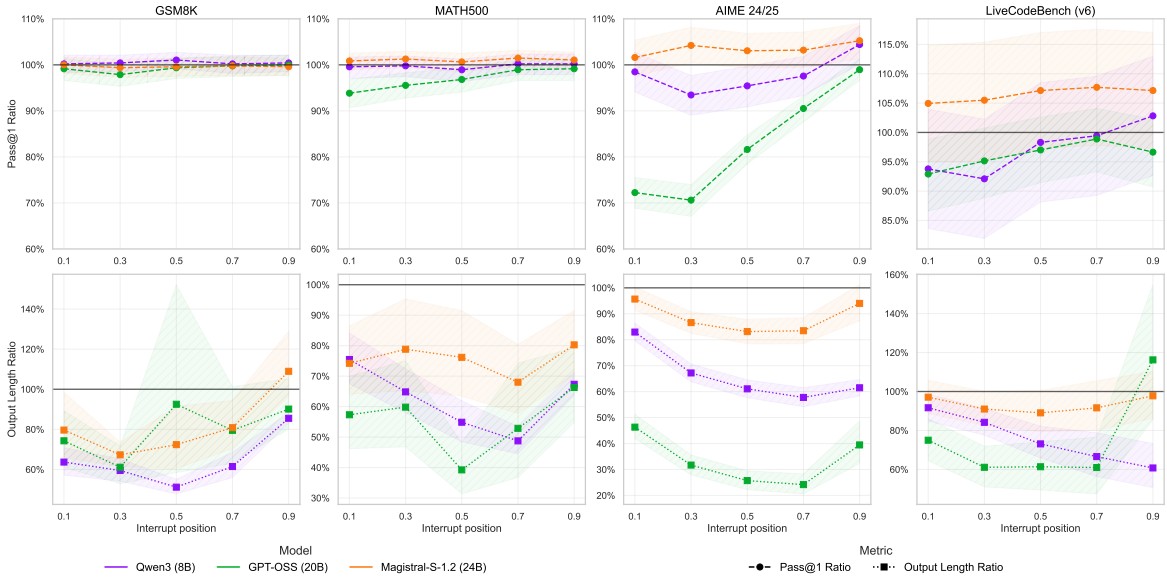

*Figure F.23.* **Efficiency and Accuracy under Soft Interrupts.** (Top) Performance under soft interrupts is comparable to that of full thinking, with the exception of hard AIME problems. (Bottom) Models generally adhere to speedup instructions, with total output lengths shorter than that of full thinking (i.e., ratio is less than 1).

## F.6. Update-Driven Interrupt with Speedup

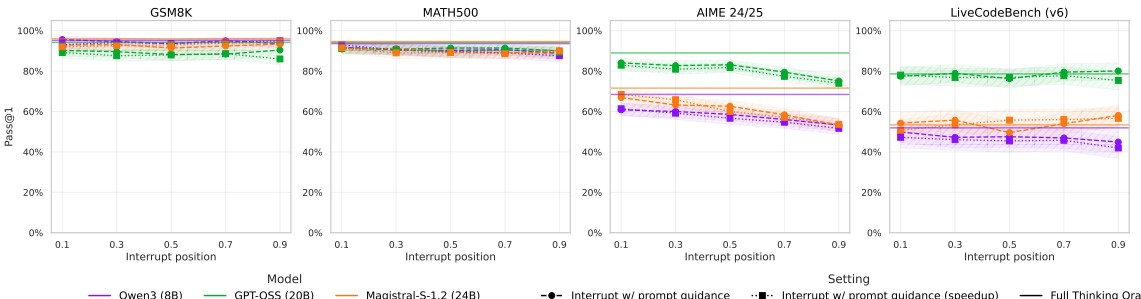

*Figure F.24.* **Accuracy under Update-Driven Speedup Interrupts.** There are no significant differences between regular and speedup update-driven interrupts.

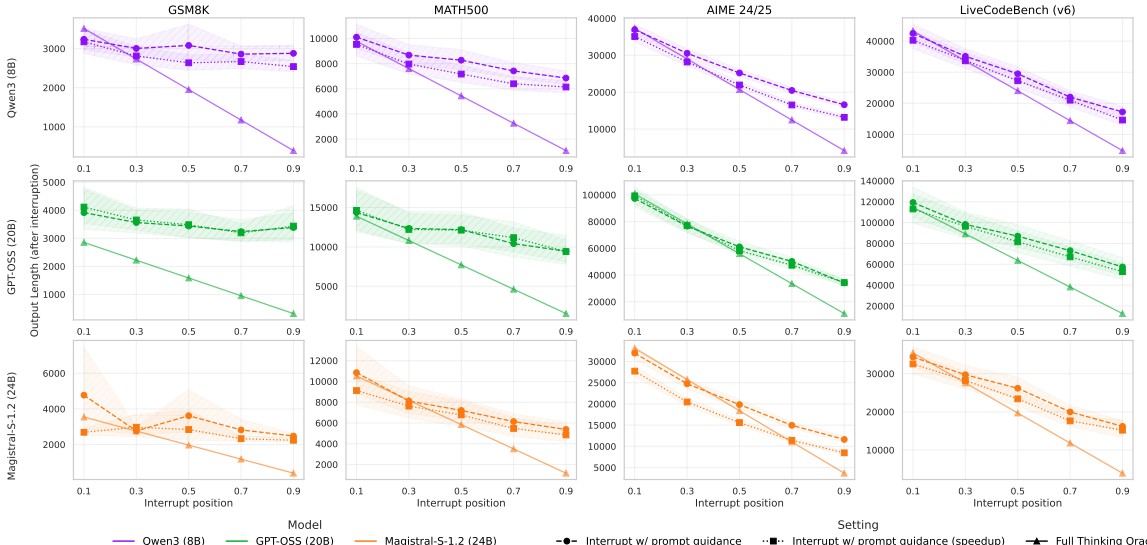

*Figure F.25.* **Output Length under Update-Driven Speedup Interrupts.** Compared to regular update-driven interrupts, additional speedup instructions can lower token usage for specific scenarios (e.g., Magistral-S-1.2 on AIME-24/25), while having similar accuracy (see Fig. F.24).

## F.7. Additional Scaling Experiments

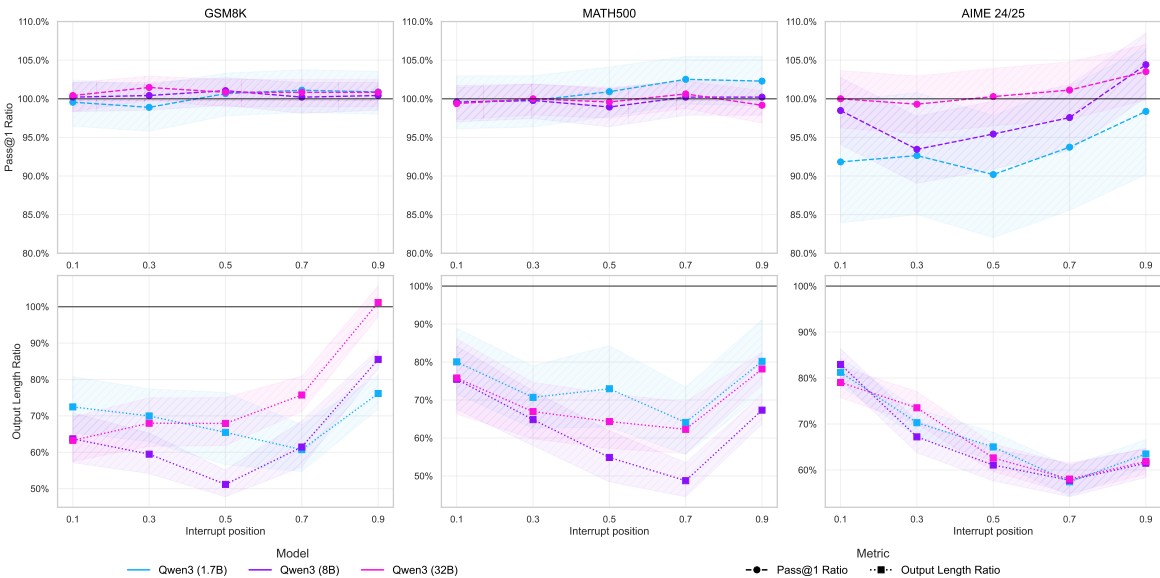

*Figure F.26.* **Efficiency and Accuracy under Soft Interrupts by Model Scale.** (Top) Models mostly preserve their original accuracy under soft interrupts, with the exception of the smallest model (Qwen3-1.7B) on AIME-24/25. (Bottom) Models generally adhere to speedup instructions, with total output lengths shorter than that of full thinking (i.e., ratio is less than 1), with no significant differences across models.

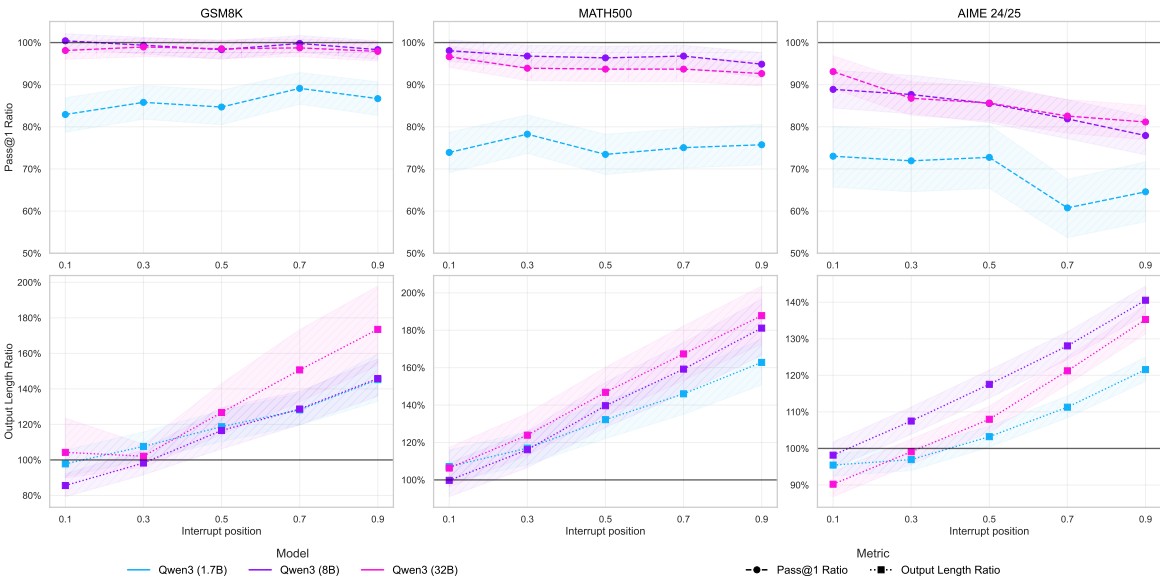

*Figure F.27.* **Efficiency and Accuracy under Update-Driven Interrupts by Model Scale.** (Top) On easier problems (GSM8K and MATH500), larger models preserve their original accuracy, while Qwen3-1.7B model drops its original accuracy by up to 75%. On AIME problems, the performance of the larger models is degraded as well. (Bottom) All models increase their thinking token usage after the update interrupt, with the largest model (Qwen3-32B) showing the greatest increase on the AIME dataset.

