# OpenReview forum: "Are Large Reasoning Models Interruptible?"
_ICML.cc/2026/Conference — ICML 2026 regular_

### Official Review · Reviewer_Ttba · 2026-02-27

**Soundness:** 3
**Presentation:** 3
**Significance:** 2
**Originality:** 2
**Overall Recommendation:** 4
**Confidence:** 4

**Summary:**

The paper investigates a setting in which large reasoning models (LRMs) are intervened upon during their reasoning process. Specifically, the authors consider two types of interventions: (1) hastening the model to produce its final answer earlier than it otherwise would, and (2) modifying the context mid-reasoning so that a different final answer becomes appropriate. They study how such “interruptions,” applied at various points along the reasoning trace, affect model performance in terms of answer accuracy and response length and name several modes of failure caused by such interruptions.

**Compliance With Llm Reviewing Policy:**

Affirmed.

**Final Justification:**

I thank the authors for addressing my concerns and raise my score to 4.

**Key Questions For Authors:**

**Q1:** Would the observed degradations persist if the interruption mechanism were more structured or adaptive?
A positive result here would clarify whether the brittleness is inherent to interruption or specific to the chosen mechanism.

**Q2:** Prompt guidance partially mitigates self-doubt on easier tasks. Beyond this method could you think of a more robust way to fix this pathology or avoid panic/reasoning leakage?

**Q3:** How would you translate your relative-position protocol into a deployable interruption policy?

**Limitations:**

yes

**Strengths And Weaknesses:**

**Strengths**
* The problem setting is practically compelling. Evaluating LRMs under (i) time-constrained interruptions (for efficiency/latency) and (ii) update-driven mid-inference context changes (for dynamic environments) directly matches real end-user workflows (wanting answers sooner, or modifying task beliefs/specifications midstream). This “dynamic world” framing is a meaningful direction in the area of LRM research.

* The experimental coverage is strong: multiple model families, and datasets that are appropriate for eliciting long-form reasoning (math + coding) at nontrivial difficulty. The use of repeated trials (e.g., multiple runs for AIME-style high-variance settings) and reporting confidence intervals improves credibility and helps separate genuine effects from noise.

* The paper is clearly written and well-structured. The figures and experimental breakdown communicate the three pathologies distinctly (answer-length inflation under leakage, accuracy degradation attributed to panic vs. self-doubt which is mainly credited to the model at test).

* The formalization of interruption as a two-stage process (pre-interrupt trace, then post-interrupt continuation conditioned on an intervention token/message) is clean and supports reproducible measurement of both accuracy under intervention and post-interrupt compute (via length). The methodology of labeling and quantifying failure modes (leakage/panic/self-doubt) also supports the paper’s analytical soundness.

**Weaknesses**
* The interruption mechanism feels somewhat weak/limited: fixed-ratio truncations plus generic “end thinking / force answer / answer faster” style prompts may not reflect best-known ways to reduce reasoning cost while preserving quality. Several recent lines of work show you can compress or bypass verbose reasoning with minimal loss, and sometimes better performance, via more structured inference-time controls [1,2,3]. Implication: If “better” interruption/control baselines were used, the absolute degradation numbers (and possibly the discovered failure frequencies) might change, and the paper’s conclusions could be reframed as “naive interrupts are harmful” rather than “interruptibility is intrinsically brittle.”

* As the authors also note, the current paper is mostly diagnostic. It’s hard to translate the findings into practice without mechanisms to (a) detect when an interruption is likely harmful or (b) choose an interruption strategy (timing + content) that preserves quality. If the paper proposed even a lightweight decision rule (e.g., heuristics for identifying self-doubt loops or instability and adapting the interrupt content accordingly), it could move beyond “analysis-only” and become more actionable.

* The protocol “generate the full trace, then simulate interruption at X% of RT” is reproducible, but it differs from real-time interruption: in a live system you typically don’t know “where you are” in the model’s reasoning, nor can you reliably map to a fraction of the eventual trace length. This further limits the translation of the evaluation protocol into deployed interruption policies, and also raises the question of whether online interrupts induce the same internal dynamics as post-hoc splicing.

* The update-driven dataset updates may be too “soft” to fully test strategic adaptation. Many described updates resemble literal/value changes (math variable values; code constraints/names/starter code tweaks) rather than requiring the model to substantially revise its approach or choose a different reasoning plan. This could under-measure failures that arise when the update demands a genuine change in solution strategy (e.g., switching algorithms, changing proof approach, or abandoning a partially built plan).



[1] Dinardi, R. C., Yamamoto, B., Costa, A. H. R., & Jordao, A. (2025). The virtues of brevity: Avoid overthinking in parallel test-time reasoning. In NeurIPS 2025 Workshop on Efficient Reasoning.

[2] Li, Z., Li, J., Jiang, G., Song, L., Lian, D., & Wei, Y. (2026). Scaling reasoning hop exposes weaknesses: Demystifying and improving hop generalization in large language models. In Proceedings of the International Conference on Learning Representations (ICLR 2026).

[3] Li, S., Huang, N., Saxena, N., Luo, N., Lin, V., Zhu, K., & Dev, S. (2025). Inference-time chain-of-thought pruning with latent informativeness signals. In NeurIPS 2025 Workshop on Efficient Reasoning.

---

> ### Author Rebuttal · Authors · 2026-03-31
>
> We appreciate the reviewer’s detailed comments and suggestions. We address the reviewer’s concerns below:
>
> ---
> **W1, Q1: Budget-aware methods.** Our primary goal in this work is not to develop an optimal interrupt mechanism but to study models’ behaviors and pathologies under user-provided dynamic interrupts. This is complementary to the work on reasoning compression or inference-time control [1-3]; methods that shorten reasoning may reduce the frequency of interruption, but they do not *directly* address robustness to interruptions. To demonstrate, we ran Chain-of-Draft (CoD) [4] with Qwen3-8B. The results are provided below:
>
> - Hard interrupt (accuracy, final answer length): https://anonymous.4open.science/r/lrm82EF/hardcod.png
> - Soft interrupt (accuracy): https://anonymous.4open.science/r/lrm82EF/softcod.png
> - Soft interrupt (output length): https://anonymous.4open.science/r/lrm82EF/softcodlen.png
> - Update-driven interrupt (accuracy): https://anonymous.4open.science/r/lrm82EF/updcod.png
>
> While CoD often reduces output length, it does not eliminate the three interruption pathologies.
>
> 1. **Hard interruption:** CoD reduces the number of output tokens, but reasoning leakage still remains severe. For example, when interrupted at 0.3, answer lengths are still inflated by 1.38 and 6.27 times compared to the no-interruption scenario on the AIME and LCB-v6 datasets, respectively.
> 2. **Soft interruption:** Under speed-up instructions, the CoD variant still exhibits substantial panic behavior, compared to the regular assistant-turn interrupt (13.1% vs 3.8% at interrupt position 0.3 with panic threshold 1% on AIME), and its accuracy is degrades more than the regular reasoning baseline.
> 3. **Update-driven interruption.** Both methods exhibit self-doubt behavior on AIME and LCB-v6, which is partially mitigated through prompt guidance. The CoD variant achieves slightly lower accuracy across interrupt positions.
>
> It is thus important to introduce interruptible reasoning/update-driven interaction as an evaluation setting beyond just the efficiency gains. This broader view, treating interruption as an interaction setting rather than only a budget-control mechanism, is what distinguishes our paper.
>
> [4] Xu, S., Xie, W., Zhao, L., and He, P. Chain of draft: Thinking faster by writing less. arXiv. 2025.
>
> ---
> **W2, Q2: Justification of actionability and potential fixes.** While defining and evaluating the problem is a necessary first step toward developing effective solutions, in this work, we further make the problem actionable beyond benchmark evaluation in two ways. We both characterize three recurring failure taxonomies associated with three practically motivated interruption settings (highlighted by reviewer `gggx`), and provide a reasonable and actionable lower-bound for future work through our prompt-guidance baseline (also discussed by reviewer `3tXH`).
>
> We believe that a robust long-term solution would likely require interruption-aware training. One value of our taxonomy is precisely that it makes such training objectives more concrete: models can be exposed to forced answering, speed-up requests, and mid-reasoning updates, while being optimized to reduce leakage, panic, and self-doubt. Overall, we believe the problem setup, benchmark, prompt-guidance baseline, and failure taxonomy provide a concrete foundation for future research.
>
> ---
> **W3, Q3: Relative position interrupt clarification.** To clarify, in this work, we do not suggest that models should be interrupted at X% in the reasoning trace; we instead focus on **showing** how models respond to such interruptions. Thus, we could simulate interruptions after either an absolute token number or after a fixed percentage of the full trace. We choose relative position, as it provides a reproducible way to compare interruption timing across tasks and models with different verbosity levels. We discuss this motivation further, and provide absolute-token-level interrupt analysis in Appendix E.3. It is true that relative-position interruption is impractical, but this method is not meant to prescribe a production policy; it serves as a normalization tool in the analysis.
>
> ---
> **W4: Update-driven interrupt setup.** Update-driven interruption is practically motivated by realistic user interactions (e.g., variable changes, spec updates), while the model is still reasoning. Such updates challenge current LRMs: even under this setting, models show clear self-doubt behavior and substantial performance degradation across several model families. We view this paper as a first step toward a broader study of update-driven interruption. It provides a concrete formulation of the problem, a scalable data construction pipeline, and a failure taxonomy that future work can build on directly. As discussed by the reviewer and in the limitations section, there are several natural extensions. We will add this more explicitly to the limitations and future work discussions in the final revision.

---

> > ### Author Rebuttal · Reviewer_Ttba · 2026-04-03
> >
> > I appreciate the inclusion of results with respect to CoD and thank the authors for the detailed answers. I remain concerned about the significance of the work. The authors briefly state in the Conclusion section that they hope this work will serve as a foundation to train models that are robust under more dynamic interaction, however they do not provide tools or suggestions to deal with panic, self-doubt, or reasoning-leakage both after and during training.

---

> > > ### Author Response · Authors · 2026-04-03
> > >
> > > We thank the reviewer for the quick follow-up. We agree that the ultimate goal is not only to diagnose failure modes, but also to train models that are robust to different types of interrupts. We argue that this paper contributes the prerequisite infrastructure for that next step: a clear problem formulation, a benchmark, and measurable failure metrics. More concretely, we would like to clarify how our work and findings support future efforts in developing interruption-robust LRMs:
> > >
> > > **Problem formulation and evaluation.** Robustness to interruption is a critical property for LRMs deployed in real-world interactive settings, yet it remains largely understudied (noted by Reviewers such as `Bege`, `3tXH`). Before the field can train LRMs for dynamic, interruptible interaction, it first needs (1) a concrete setting to optimize for, (2) a reliable benchmark to measure progress, and (3) a diagnosis of current failure modes. Our paper provides all three. We formalize interruptible reasoning, construct a human-verified benchmark, evaluate five state-of-the-art LRM families, and identify three recurring pathologies — reasoning leakage, panic, and self-doubt — that are largely invisible under standard static benchmarks. We also open-source our benchmark construction pipeline, enabling easy adaptation beyond math and coding.
> > >
> > > **Enabling training.** Our results reveal a clear gap in how current LRMs are trained. We believe a natural next step is post-training. A straightforward RL extension is to inject simulated user interruptions during rollout, including both time-constrained and update-driven interrupts, and train the model to continue adaptively from the interrupted state. The reward can combine answer correctness with explicit penalties for:
> > >
> > > - Reasoning leakage: leaking that reasoning into the final answer when the model has been asked to stop thinking,
> > > - Panic: terminating too abruptly after a speed-up request and therefore hurting correctness,
> > > - Self-doubt: failing to incorporate user updates or unnecessarily abandoning an otherwise valid reasoning trajectory.
> > >
> > > Under this view, the paper does not only diagnose failures; it also suggests a concrete direction for training through the proposed interruption mechanisms and failure metrics. We hope this clarification addresses the reviewer's concern, and we will revise the paper to make the significance of the work for future directions more explicit. We thank the reviewer again for the thoughtful feedback.

---

### Official Review · Reviewer_gggx · 2026-03-02

**Soundness:** 3
**Presentation:** 4
**Significance:** 4
**Originality:** 3
**Overall Recommendation:** 5
**Confidence:** 4

**Summary:**

The paper investigates the robustness and behavior of Large Reasoning Models (LRMs) when exposed to dynamic mid-inference environments. Moving beyond the traditional "frozen world" static evaluation paradigm, the authors introduce a public evaluation suite across math and programming benchmarks designed to test models under two realistic scenarios: time-constrained interruptions (hard stops and soft speedup requests) and update-driven interruptions (where task specifications change mid-reasoning). The study reveals that static evaluations consistently overestimate LRM robustness. The authors identify three novel and recurring pathologies: (1) reasoning leakage, where models continue their reasoning process within the final answer block after a hard interruption; (2) panic, where models prematurely abandon reasoning and return incorrect answers when instructed to speed up; and (3) self-doubt, where models fail to successfully incorporate updated information and instead second-guess their prior steps. The paper also demonstrates that basic prompt guidance can partially mitigate update-driven failures, though the issue remains largely unsolved, particularly for harder problems and smaller models.

**Compliance With Llm Reviewing Policy:**

Affirmed.

**Final Justification:**

Strengths：

1.The empirical evaluation is rigorously designed, covering a diverse set of state-of-the-art open-weight model families and utilizing well-established benchmarks. The categorization of interventions into time-constraint and update-driven is logical. The identified pathologies are well-supported by both quantitative metrics and detailed qualitative examples.

2.The identified failure modes reasoning leakage, panic, self-doubt are novel and actionable.The evaluation methodology is thorough across different axes. Evaluating varying interruption points provides a granular look at how temporal dynamics affect model recovery and accuracy.

Weakness:

1.The definition of the "panic" metric is somewhat arbitrary. The paper defines panic as the model closing its thinking after using less than 1% of its left context limit. There is no justification provided for why 1% is the correct threshold. A sensitivity analysis showing how the panic rate fluctuates with different thresholds (e.g., 2% or 5%) is necessary to ensure the metric is robust.

2.While the figures are informative, the complete absence of results tables in the main text makes it difficult to extract precise baseline numbers for future comparison. Adding at least one summary table of absolute pass rates and error mode percentages would improve reproducibility and readability.

**I believe the authors' rebuttal has effectively addressed my concerns. Therefore, I maintain my rating of 5 for this submission.**

**Key Questions For Authors:**

1.How sensitive are the “panic” and “self-doubt” rates to their detection thresholds (e.g., 1% remaining-context rule; classifier confidence)?

2.In Figure 2, what does the underlying distribution of the answer lengths look like? Are the large error bars driven by a small number of extreme outliers, or is the leakage consistently high across most samples?

3.The authors argue that math/coding tasks are chosen because they induce long reasoning traces and allow reproducible "time" control. But can you provide the results on the QA datasets?

**Limitations:**

yes

**Strengths And Weaknesses:**

Strengths：

1.The empirical evaluation is rigorously designed, covering a diverse set of state-of-the-art open-weight model families and utilizing well-established benchmarks. The categorization of interventions into time-constraint and update-driven is logical. The identified pathologies are well-supported by both quantitative metrics and detailed qualitative examples.

2.The identified failure modes reasoning leakage, panic, self-doubt are novel and actionable.The evaluation methodology is thorough across different axes. Evaluating varying interruption points provides a granular look at how temporal dynamics affect model recovery and accuracy.

Weakness:

1.The definition of the "panic" metric is somewhat arbitrary. The paper defines panic as the model closing its thinking after using less than 1% of its left context limit. There is no justification provided for why 1% is the correct threshold. A sensitivity analysis showing how the panic rate fluctuates with different thresholds (e.g., 2% or 5%) is necessary to ensure the metric is robust.

2.While the figures are informative, the complete absence of results tables in the main text makes it difficult to extract precise baseline numbers for future comparison. Adding at least one summary table of absolute pass rates and error mode percentages would improve reproducibility and readability.

---

> ### Author Rebuttal · Authors · 2026-03-31
>
> We appreciate the reviewer’s detailed comments and suggestions. We address the reviewer’s concerns below:
>
> ---
>
> **W1, Q1 (1): Threshold Sensitivity Analysis and Panic Classifier Validation.** If the model failed after a soft (speedup) interrupt and stopped its reasoning within 1% of its remaining context, we classify it as a “panic” behavior. In Figure 1(b), we use 1% as a threshold because, in many failure cases, the model stops reasoning almost immediately after a speed-up prompt, often with statements like “We have all the elements needed for the final answer…” (Lines 850–855). The 1% rule was chosen to capture these abrupt panic terminations.
>
> To test sensitivity, we select GPT-OSS-20B (model with the highest panic rate) and measure panic error rates across all interruption positions on AIME and LiveCodeBench-v6 (LCB-v6) under multiple thresholds. The results are stable, with an average coefficient of variation of 1.8% on AIME and 4.8% on LCB-v6. We also manually labeled 50 samples from the assistant turn soft interrupt experiment data at 0.3 interruption position and found 90% agreement between human judgment and the 1% heuristic. The results are provided below.
>
> **A. Panic Analyses on AIME 24/25**
>
> | Interrupt Position | **Panic @ 0.5%** | **Panic @ 1%** | **Panic @ 1.5%** | **Panic @ 2%** | Coefficient of Variation (CV) |
> | --- | --- | --- | --- | --- | --- |
> | 0.1 | 27.1% | 27.2% | 27.2% | 27.3% | 0.3% |
> | 0.3 | 30.0% | 30.5% | 30.5% | 30.5% | 0.8% |
> | 0.5 | 18.2% | 18.8% | 18.9% | 18.9% | 1.8% |
> | 0.7 | 10.7% | 11.1% | 11.4% | 11.4% | 3.0% |
> | 0.9 | 4.5% | 4.6% | 4.8% | 4.8% | 3.2% |
>
> **B. Panic Analyses on LiveCodeBench-v6**
>
> | Interrupt Position | **Panic @ 0.5%** | **Panic @ 1%** | **Panic @ 1.5%** | **Panic @ 2%** | Coefficient of Variation (CV) |
> | --- | --- | --- | --- | --- | --- |
> | 0.1 | 5.0% | 5.0% | 5.8% | 5.8% | 8.6% |
> | 0.3 | 8.3% | 8.3% | 8.3% | 8.3% | 0.0% |
> | 0.5 | 5.6% | 7.1% | 7.1% | 7.1% | 11.2% |
> | 0.7 | 6.0% | 6.0% | 6.0% | 6.0% | 0.0% |
> | 0.9 | 4.9% | 4.9% | 4.9% | 5.3% | 4.0% |
>
> ---
>
> **W1, Q1 (2): Self-doubt classifier validation.** The self-doubt phenomenon is typically marked by repeated uncertainty markers, such as multiple "wait"-like words or explicit second-guessing right after the update-driven interrupts. This behavior is, thus, easier to capture with automated annotation. To verify the LLM-based classifier’s outputs, the authors of the paper manually labeled 50 random samples from the update-driven experiment data at 0.3 interruption position and found 88% agreement rate with the GPT-5-based classifier.
>
> ---
>
> **W2: Suggestion to add results tables.** Thanks for the suggestion. We will add summary tables in the final revision.
>
> ---
>
> **Q2: Answer-length distribution in Figure 2.** We examined the underlying answer-length distribution across models, datasets, and interrupt positions and found that the large error bars and significant differences between the mean values are not driven by a small number of extreme outliers. The final answer length distribution under the hard interrupt mechanism on LiveCodeBench-v6 at 0.1 interruption position is provided under this anonymous link:
>
> https://anonymous.4open.science/r/lrm82EF/dist.png
>
> Under hard interruption, the distribution shifts upward substantially, especially in the median and upper quartiles. This shows that reasoning leakage occurs across a large fraction of samples rather than only a few anomalous cases. For example, for Qwen3-8B, the median answer length changes only modestly (1.6k → 1.7k), with a significant increase in the 75th percentile (Q3: 3.1k → 32.7k), showing that leakage is concentrated in a substantial subset rather than a handful of isolated outliers. For GPT-OSS-20B and Magistral-S-1.2-24B, the effect is even clearer across the distribution.
>
> Overall, these statistics suggest that the leakage behavior is clearly not an artifact of only a few outliers. We will add these distributional statistics to the revision for clarity.
>
> ---
>
> **Q3: Extension to QA tasks.** We agree that extending the study to QA datasets would be valuable. However, most conventional QA tasks, particularly short-form or multiple-choice formats, produce reasoning traces that are too brief to make interruption practically motivated or even well-defined: if the model resolves the question in a few hundred tokens, there is little meaningful 'mid-reasoning' context to interrupt. Beyond trace length, QA tasks also typically lack the unambiguous, auto-gradable ground truth that math and coding provide, making it significantly harder to isolate whether post-interruption degradation reflects a failure of reasoning continuity versus evaluation noise. That said, we agree that long-horizon QA settings such as deep-research or multi-hop QA are promising extensions where interruption would be both practical and impactful, and we will add this as an explicit future direction in the final revision.

---

> > ### Author Rebuttal · Reviewer_gggx · 2026-04-01
> >
> > Thank you for the detailed response.I will maintain my score.

---

### Official Review · Reviewer_3tXH · 2026-03-12

**Soundness:** 3
**Presentation:** 3
**Significance:** 2
**Originality:** 3
**Overall Recommendation:** 4
**Confidence:** 4

**Summary:**

This paper introduces interruptible reasoning*as an evaluation setting for large reasoning models, focusing on two practically motivated interruption types of time-constrained interrupts and update-driven interrupts . It reports systematic experiments on math and coding benchmarks like GSM8K, MATH500, AIME 24/25, LiveCodeBench v6, and identifies three recurring failure modes of reasoning leakage, panic, and self-doubt. As a simple mitigation, it proposes a short prompt-guidance postfix that can reduce self-doubt on easier tasks.

**Compliance With Llm Reviewing Policy:**

Affirmed.

**Final Justification:**

The authors' rebuttal resolved my concerns and I'm raising my score / confidence to 4.

**Key Questions For Authors:**

How do strong **anytime/budget-aware methods** (including representative approaches from your own related work list) perform under your interruption and update-driven protocols?

Can you include a best-effort **proxy** for interruption/updates using stop-and-resume prompting (streaming + truncation + re-injection), and report how conclusions change?

Can you extend evaluation to more industry-representative reasoning systems or at least provide evidence that the observed effects persist at the high end (beyond the current open-weight set)?

**Limitations:**

Yes

**Strengths And Weaknesses:**

### Strengths

**(1) The problem is practical and clearly defined, and the taxonomy is sharp.** The paper formalizes a real deployment pain point (mid-inference interruption and mid-stream updates) and names the observed pathologies in a way that is easy to reason about and measure (leakage, panic, self-doubt).

**(2) It provides a simple, actionable baseline.** The prompt-guidance postfix is a lightweight, training-free intervention that improves update-driven behavior on easier tasks, which makes the diagnosis more actionable than pure characterization.

### Weaknesses

**(1) Missing strong baselines under the proposed protocol.** The related work section itself discusses closely aligned inference-time control methods (e.g., budget forcing, NoThinking, ThinkLess, Chain of Draft, Constrained-CoT), but the experimental section does not compare against representative anytime/early-exit/budget-aware baselines within the same interruption protocol.
This makes it hard to conclude whether the reported degradation is intrinsic “non-interruptibility,” versus an artifact of using naive interruption strategies rather than state-of-the-art control.

**(2) The “not feasible” justification for API-based evaluation is not fully convincing without best-effort proxies.** Not evaluating closed-source LRMs is understandable, but the paper could still provide proxy evaluations that approximate interruption and updates using standard streaming/stop-and-resume patterns, or at least establish upper/lower bounds. Without this, the external validity to widely deployed systems remains unclear.

**(3) Model coverage is skewed toward smaller open-weight systems, leaving “real business needs” under-tested.** The main analysis centers on **Qwen3-8B, GPT-OSS-20B, Magistral-S-1.2-24B**, with additional results for **DeepSeek-R1-Llama3 (8B/70B)** and **Nemotron-3-Nano-30B A3B** in the appendix.
While the small-/mid-scale results are informative and the observed behaviors are plausible, the paper does not convincingly demonstrate performance under the operational constraints and model families that dominate real deployments. That gap limits the paper’s significance.

---

> ### Author Rebuttal · Authors · 2026-03-31
>
> We are grateful for the reviewer’s valuable feedback, and below we address their main concerns.
>
> ---
>
> **W1, Q1: Missing Strong Baselines.** We agree that including interruption of inference-time control baselines would strengthen the paper. Thus, we ran Chain-of-Draft (CoD) with Qwen3-8B. The results are provided below:
>
> - Hard interrupt (accuracy, final answer length): https://anonymous.4open.science/r/lrm82EF/hardcod.png
> - Soft interrupt (accuracy): https://anonymous.4open.science/r/lrm82EF/softcod.png
> - Soft interrupt (output length): https://anonymous.4open.science/r/lrm82EF/softcodlen.png
> - Update-driven interrupt (accuracy): https://anonymous.4open.science/r/lrm82EF/updcod.png
>
> While CoD often reduces output length, it does not eliminate the three interruption pathologies.
>
> 1. **Hard interruption:** CoD reduces the number of output tokens, but reasoning leakage still remains severe. For example, when interrupted at 0.3, answer lengths are still inflated by 1.38 and 6.27 times compared to the no-interruption scenario on the AIME and LCB-v6 datasets, respectively.
> 2. **Soft interruption:** Under speed-up instructions, the CoD variant still exhibits substantial panic behavior, compared to the regular assistant-turn interrupt (13.1% vs 3.8% at interrupt position 0.3 with panic threshold 1% on AIME), and its accuracy is degrades more than the regular reasoning baseline.
> 3. **Update-driven interruption.** Both methods exhibit self-doubt behavior on AIME and LCB-v6, which is partially mitigated through prompt guidance. The CoD variant achieves slightly lower accuracy across interrupt positions.
>
> Such baselines are, however, orthogonal to our main goal, as inference control mainly aims to reduce reasoning cost, whereas interruption additionally can inject new information mid-reasoning. Here, we thus introduce interruptible reasoning as an evaluation setting, formalize two practically motivated interruption types, and characterize three resulting failure modes. This broader view, treating interruption as an interaction setting rather than only a budget-control mechanism, is what distinguishes our paper.
>
> ---
>
> **W2, Q2: API-based evaluation.** API-based evaluation is infeasible because model providers do not expose the full reasoning trajectory. For example, Grok exposes only encrypted reasoning, and OpenAI provides reasoning summaries rather than raw traces. When the reasoning trace is not fully observable and editable, any experiment becomes a proxy for true interruption. For this reason, we focus on open models.
>
> Despite this limitation, we conducted a best-effort proxy experiment with GPT-5.4-Mini by disabling internal reasoning and instead prompting the model to produce an explicit chain-of-thought in its answer, with separate “thinking” and “answer” fields. Because this setup disables the model’s native reasoning mode, overall performance is substantially below the model’s true capability. The results are below:
>
> - Hard interrupt (accuracy, final answer length): https://anonymous.4open.science/r/lrm82EF/hard54.png
> - Soft interrupt (accuracy): https://anonymous.4open.science/r/lrm82EF/soft54.png
> - Soft interrupt (output length): https://anonymous.4open.science/r/lrm82EF/soft54len.png
> - Update-driven interrupt (accuracy): https://anonymous.4open.science/r/lrm82EF/upd54.png
>
> All three pathologies emerge under this proxy setting, suggesting they are structural failure modes of interrupted reasoning rather than artifacts of a particular model family or reasoning format. We hope this additional experiment helps address the reviewer’s concern, while also clarifying why a fully faithful reproduction on proprietary APIs is currently not possible.
>
> ---
>
> **W3, Q3: Skewed toward smaller open-weight systems.** While the main tables focus on small- to mid-scale open-weight models, our goal was to first establish the phenomenon in settings where interruption can be easily implemented and reproduced. The appendix already includes a 70B-scale model (where the same pathologies emerge), and following the reviewer’s suggestion, we further evaluated GPT-OSS-120B. The results are below:
>
> - Hard interrupt (accuracy): https://anonymous.4open.science/r/lrm82EF/hardoss.png
> - Soft interrupt (accuracy): https://anonymous.4open.science/r/lrm82EF/softoss.png
> - Soft interrupt (output length): https://anonymous.4open.science/r/lrm82EF/softosslen.png
> - Update-driven interrupt (accuracy): https://anonymous.4open.science/r/lrm82EF/updoss.png
>
> We observe the same three pathologies at this larger scale as well. Consistent with Sec. 6.1 and Appendix E.6, larger models have (1) higher panic rates, 30.7% (OSS 20B) vs 52.8% (OSS 120B), (2) less reasoning leakage, and (3) less sensitivity to the guidance text in update-driven interrupt settings. Our results, now across 9 models and two within-family scaling comparisons, clearly demonstrate that the observed phenomena are not isolated occurrences.

---

> > ### Author Rebuttal · Reviewer_3tXH · 2026-04-02
> >
> > I thank the authors for their detailed rebuttal. I'm raising my score to 4.

---

### Official Review · Reviewer_Bege · 2026-03-13

**Soundness:** 2
**Presentation:** 3
**Significance:** 3
**Originality:** 3
**Overall Recommendation:** 4
**Confidence:** 4

**Summary:**

This work studies whether large reasoning models remain reliable when the “frozen-world” assumption is broken. The authors formalize interruptible reasoning under three settings: hard interruption, where reasoning is terminated and the model must answer immediately; soft interruption, where the model is asked to speed up; and update-driven interruption, where new information changes the task specification mid-reasoning. They define interruption-conditioned accuracy and post-interruption output length as evaluation metrics, build an interrupt benchmark on top of GSM8K, MATH500, AIME24/25, and LiveCodeBench-v6, and evaluate several open-weight reasoning models. The main findings are that static evaluation overestimates robustness in dynamic settings, and that models exhibit recurring failure modes such as reasoning leakage, panic, and self-doubt.

**Compliance With Llm Reviewing Policy:**

Affirmed.

**Key Questions For Authors:**

1. For “panic” and “self-doubt,” can the authors report human evaluationon a sample, as well as sensitivity to the panic threshold?

2. How sensitive are the prompt-guidance gains, to paraphrases of the injected update and guidance text?

3. How many update-driven examples were accepted as generated, how many were manually rewritten, and what was the human verification protocol?

**Limitations:**

Yes

**Strengths And Weaknesses:**

Strengths:

1. This is a well-motivated work. The question is interesting: current reasoning-model evaluation is overwhelmingly static, while many real deployments are interactive and require interruption, acceleration, or mid-course correction.

2. The paper is written with exceptional clarity. The method, experimental setup, and results are described in sufficient detail to facilitate understanding.

Weaknesses:

1. The contribution is mainly about benchmark + analysis + taxonomy, with only a very light mitigation baseline, which limits the depth compared with the other interruption-aware works.

2. The external validity is limited. The main experiments intervene inside the assistant turn, partly because some models do not reliably support user-turn interruption mid-thinking.

3. Some of the analyses depend on model-based labeling. “Panic” is defined via a threshold on how much of the remaining context is used after a soft interrupt, and “self-doubt” is identified using a GPT-5-based classifier with only qualitative validation mentioned.

---

> ### Author Rebuttal · Authors · 2026-03-31
>
> We are grateful for the reviewer’s valuable feedback. Below, we address their main concerns.
>
> ---
>
> **W1: Paper is primarily benchmark + analysis + taxonomy.** We believe that clearly defining the problem is a necessary precondition for finding a strong solution. Thus, in this paper, we focused on (1) formalizing the problem setup, (2) developing a benchmark with dynamic interruptions, and (3) analyzing interruption behaviors and failure modes.
>
> While prior work has mainly studied interruption from an efficiency perspective, our work instead frames interruption as a core interaction paradigm for long-horizon tasks. To the best of our knowledge, this is the first work to systematically study both time-constrained and update-driven interruption scenarios. Our results further show that many SOTA models do not reliably support user-turn interruption, and often degrade under assistant-turn settings, suggesting that this capability is not yet a standard target in current post-training. We intentionally keep our prompt guidance baseline simple for diagnosis purpose: testing if straightforward prompting already resolves the problem. Since they help only partially, we believe our benchmark, scenarios, and failure taxonomies provide a useful foundation for future post-training research (e.g., adapting them to reward functions in RL).
>
> ---
>
> **W2: Discussion on User-turn Interruptions.** We agree that user-turn interruption is an important interruption mechanism, and we analyze it in Sec. 6.2. The results show slightly worse performance than the assistant-turn interruption mechanism, and many models do not reliably follow the `<think>…</think>` template after a user-turn interruption due to a drift from the model’s learned distribution. This is why we use assistant-turn interruptions in the main analysis.
>
> ---
>
> **W3, Q1 (1): Threshold Sensitivity Analysis and Panic Classifier Validation.** We use the 1% remaining-context threshold as a heuristic because many panic failures happen when the model terminates its reasoning almost immediately after a speed-up prompt such as *"We have all the elements needed for the final answer…”* (see Lines 850-855 of the submission). To validate this choice, we analyzed GPT-OSS-20B, the model with the strongest panic behavior in our experiments, across multiple thresholds on both AIME and LiveCodeBench-v6. The resulting panic rates are highly stable, with a low average coefficient of variation across thresholds. We also manually labeled 50 samples at interruption position 0.3 and found 90% agreement between human judgment and the 1% heuristic. Please also see our response to Reviewer `gggX`’s W1, Q1 (1) for the full sensitivity analysis tables and additional details.
>
> ---
>
> **W3, Q1 (2): Self-doubt classifier validation.** The self-doubt phenomenon is typically marked by repeated uncertainty markers, such as multiple "wait"-like words or explicit second-guessing right after the update-driven interrupts. This behavior is, thus, easier to capture with automated annotation. To verify the LLM-based classifier’s outputs, the authors of the paper manually labeled 50 random samples from the update-driven experiment data at 0.3 interruption position and found 88% agreement rate with the GPT-5-based classifier.
>
> ---
>
> **Q2: Sensitivity of prompt guidance.** In the main paper, we already show that prompt guidance is effective when the injected text is written in the model’s own voice and clearly states that the update is correct and verified by the user (Lines 314–322). Figure 4 also reports both mean performance and confidence intervals, showing that the gains are meaningful across multiple models and datasets.
>
> Per the reviewer’s request, we conducted an additional prompt ablation by paraphrasing the injected guidance. The results are provided under this link:
>
> - Update-driven interrupt (accuracy): https://anonymous.4open.science/r/lrm82EF/updpara.png
>
> The difference in performance between the paraphrases of the guidance prompt is not statistically significant, suggesting that the model is not sensitive to the paraphrases of the guidance text. We will include these results in the final revision.
>
> ```python
> # The original prompt is in page 12 of the paper and here's the paraphrase.
> Truncated Reasoning Trace... The user has shared an update.
>
> <update>[PLACEHOLDER]</update>
>
> I have confirmed that this information comes directly from the user. I will now incorporate it into the current context and continue my reasoning on that basis.
> ```
>
> ---
>
> **Q3: Clarification on update-driven dataset construction.** Over 80% of the update-driven examples were accepted without changes. For human verification, we built a Gradio interface displaying the full context of each example; the human labelers (authors) manually reviewed and rewrote (if needed) every sample. While time-intensive, we believe this manual process was important for the final dataset quality.

---

> > ### Author Rebuttal · Reviewer_Bege · 2026-04-03
> >
> > Thank you for the authors' response. I have also reviewed the other reviewers' questions and the corresponding replies. At this stage, I will keep my score unchanged and reserve my final evaluation for the subsequent discussion phase.

---

### Decision · Program_Chairs · 2026-04-30

**Decision:**

Accept (regular)

**Comment:**

This paper targets the LRM's dynamic robustness against realistic settings: interruptions and dynamic context. They find that although LRMs show strong capabilities on common evaluations, they are not robust under such dynamic settings. Furthermore, they also detailed and analysed different failure cases, which could provide insights into future analysis. After the rebuttal, all the reviewers recognise the work's rigor and contributions and hold positive scores towards this paper. Therefore, I recommend accepting this paper.